# WISDOM: PROGRESSIVE CURRICULUM SYNTHESIS MAKES LLMS BETTER MATHEMATICAL REASONERS

## ABSTRACT

Large Language Models (LLMs) have demonstrated remarkable capabilities across a wide range of problem-solving tasks. Despite their success, LLMs still face significant challenges in complex reasoning, particularly with advanced mathematical problems. Existing data synthesis works either focus on the diversity of generated problems but ignore the quality of corresponding response of hard problems, or generate high-quality response of existing hard problems based on rejection sampling without synthesizing more diverse instructions. To address this gap, we introduce WISDOM, which draws inspiration from the curriculum learning and gradually synthesizes more diverse and difficult problems with high-quality responses from easy to hard based on response consistency. Based on the synthesized data, we further fine-tune and develop the WISDOM series models, achieving significant improvements across multiple mathematical reasoning benchmarks. Notably, WISDOM-7B (DSMath) achieves a score of 62.4% on MATH, matching GPT-4's performance with 2/30 correct answers on AIME2024. Furthermore, WISDOM-70B (Llama3) outperforms GPT-4 on AIME2024 with 3/30 correct answers. More data and models will be available at `https://anonymous.4open.science/r/Wisdom-math-377B`.

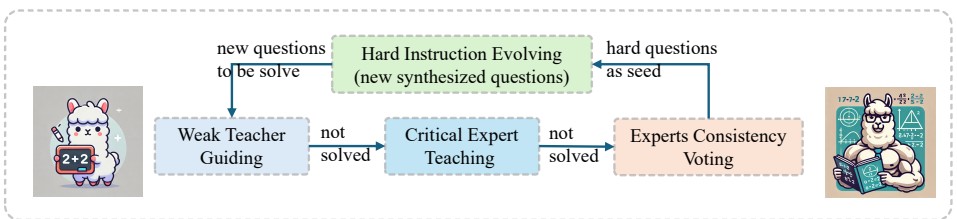

Figure 1: An overview of the construction of WISDOM through progressive curriculum synthesis.

## 1 INTRODUCTION

While large language models (LLMs) have pushed the limits of various domains (Yang et al., 2024; AI@Meta, 2024; Zhu et al., 2024), they are still struggling to handle complex reasoning tasks compared to human intelligence (Luo et al., 2023; Yu et al., 2024; Tang et al., 2024), particularly in mathematical reasoning. Ongoing research indicates that high-quality training data can notably enhance the capabilities (Chen et al., 2024; Xia et al., 2024; Liu et al., 2024a) of LLMs via instructional tuning (Chung et al., 2022). However, open-source datasets only contain a relatively low proportion of high-quality data (Liu et al., 2024b; Kang et al., 2024; Chen et al., 2024; Xia et al., 2024; Yue et al., 2024b; Li et al., 2024e). Interestingly, research shows that a carefully curated

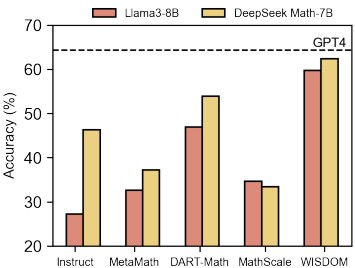

Figure 2: Comparison of LLMs fine-tuned with datasets generated by WISDOM and other synthesis methods.

small subset of data samples can yield better results than us-
ing the entire dataset (Li et al., 2024d). Nevertheless, as the scale of the pretraining increases, the availability of high-quality internet data for further instruction fine-tuning is diminishing. A practical approach to address this is through synthetic data generation. Recent studies (Liu et al., 2024b; Wang et al., 2023b; Xu et al., 2024; Li et al., 2024g) have shown that synthesizing high-quality data from raw datasets is feasible and effective for instruction tuning. While some works (Tang et al., 2024; Yu et al., 2023) focus on generating diverse problems with limited effort on the difficulty of instructions, others, i.e. DART-Math (Tong et al., 2024) generate high-quality responses of difficult problems with rejection sampling without synthesizing more diverse instructions. Meanwhile, Li et al. (2024g) devise a neuro-symbolic pipeline via auto-formalization generates mathematical problems in a domain-specific language, then "mutates" them and generates paraphrases in natural language. However, they rely on the handcrafted expressiveness of mutations to support more types of problems. Therefore, it remains a question: *how to synthesize diverse problems and high-quality responses automatically and effectively to improve the performance of LLMs on complex reasoning?*

Inspired by the curriculum learning of human, where individuals tackle complex reasoning problems by breaking them down into smaller, solvable sub-problems, we propose WISDOM. Unlike traditional curriculum learning refers to the pre-defined difficulty of tasks, we redefine it based on the model's performance. As shown in Figure 1, progressively synthesizing complex questions from simpler ones to enhance the performance of LLMs. Due to the lack of ground truth for newly synthesized questions, we innovatively use response consistency to evaluate the quality of responses and difficulty level of problems. The motivation comes from an intuitive hypothesis: simpler problems are more likely to yield consistent results across various solutions. WISDOM gradually increases problem difficulty through the following steps: *Weak Teacher Guiding*, *Critical Expert Teaching*, and *Experts Consistency Voting*, which fully leverage the inner consistency of weak models, consistency between strong and weak models, and inner consistency of strong models. Instead of generating all problems at once, we adopt a dynamic "funnel-like" filtering mechanism to evolve seed data iteratively, which progressively synthesizes new and increasingly challenging problems.

Specifically, we use MATH (Hendrycks et al., 2021) and GSM8K (Cobbe et al., 2021) as seed data and build a synthetic dataset of 1.48 million size WISDOM through a cost-effective progressive curriculum learning method from easy to hard, which encompasses problems and solutions at various levels of difficulty. Experimental results demonstrate even small language models such as Llama3-8b and DeepSeekMath-7B (DSMath-7B) can achieve competitive performance exceeding Qwen2-72B and Llama3-70B. As illustrated in Figure 2, Llama3-8B with WISDOM achieves 59.7% on MATH, which is better than Llama3-70B-instruct and existing SOTA. Additionally, the performance on out-of-domain tasks is noteworthy. Utilizing Llama3-8B, we successfully solved 17 out of 40 challenging problems, showing competitiveness with Qwen2-72B-instruct. The contributions are summarized as follows:

- We propose a novel framework for synthesizing mathematical reasoning data, which evolves the difficulty and quality of questions in a progressive manner from easy to hard. In terms of response generation, our approach is more cost-effective, and achieves remarkable SOTA on in-domain and out-of-domain tasks based on same size model.

- Compared with Rejection Sampling and Problems Rewrite, our method leverages response consistency to progressively enhance the difficulty of questions while improving response accuracy in the absence of ground truth, which is more effective for generating high-quality responses to newly synthetic questions.

- We will open-source all our models and data to drive further advancements in the open-source community for challenging mathematical reasoning tasks.

## 2 WISDOM: PROGRESSIVE CURRICULUM SYNTHESIS

### 2.1 OVERVIEW

As shown in Figure 3, we describe the workflow of our WISDOM method, which employs a curriculum learning strategy. This approach begins with the MATH and GSM8K datasets as seed data and synthesizes problems and responses of progressively increasing difficulty. The difficulty progression

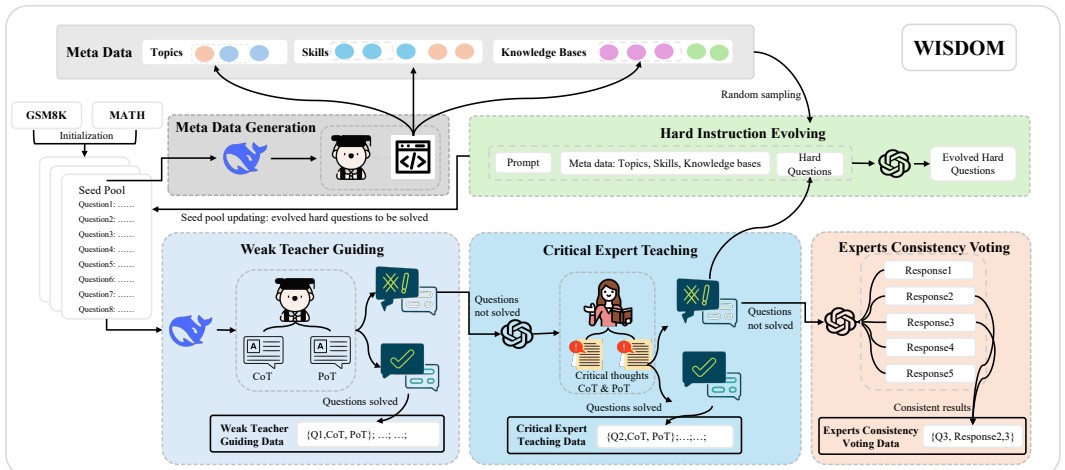

Figure 3: The overall workflow of WISDOM, which leverages Progressive Curriculum Synthesis to generate questions and responses with Deepseek Coder V2 and GPT-4o, including weak teacher guiding, critical expert teaching, experts consistency voting, and hard instruction evolving.

follows Easy to Hard Cyclic Iterative Process, a method rooted in curriculum learning principles that systematically escalates problem complexity across three distinct learning stages within each round. After five rounds of data synthesis, we obtain WISDOM ($W = S_1 \cup S_2 \cup S_3$), a high-quality, diverse dataset specifically curated for complex tasks and formatted in the style of Chain-of-Thought (CoT) datasets. S1 corresponds to the weak teacher guiding stage, S2 represents the critical expert teaching stage, and S3 reflects the experts' consistency voting stage. The data generation algorithm is described in detail in Algorithm 1.

## 2.2 DIFFICULTY-AWARE PROGRESSIVE CURRICULUM SYNTHESIS

Previous data synthesis efforts (Yu et al., 2024; Tang et al., 2024; Luo et al., 2023; Li et al., 2024a; Chan et al., 2024; Lu et al., 2024b) focused on enhancing diversity in mathematical reasoning but often overlooked question difficulty. Recent studies (Chen et al., 2024; Xia et al., 2024; Liu et al., 2024a) show that using a subset of high-difficulty data for instruction tuning enhances performance compared to using the entire dataset. To automatically define problem difficulty, we use response consistency across LLMs, including inner consistency of weak models, consistency between weak and strong models, and inner consistency of strong models. To test the hypothesis that "simpler problems can be solved by weak models with diverse solutions," we analyzed the relationship between response consistency and difficulty on 5,000 MATH dataset problems (see Appendix B.4) using the DeepSeek-V2.5 model. As shown in Table 9, results indicate a clear trend: higher difficulty correlates with lower response consistency. This enables a dynamic, "funnel-like" filtering mechanism that progressively increases problem difficulty, evolving simple questions into more challenging ones through iterative consistency evaluations.

Specifically, we start the data synthesis process with a set of seed data, which is the training set of GSM8K and MATH. Firstly, we employ Deepseek Coder V2 (DeepSeek-AI et al., 2024), the weak teacher to generate the answers $\{A_i : (c_i, p_i) : i = 1, \ldots, N_1\}$ where each answer includes both Chain of Thought (CoT) $c_i$ and Program of Thought (PoT) $p_i$. However, a weak teacher can not solve all the given questions, therefore, we filter the answers based on the CoT and PoT consistency (inner consistency). While the assumed solved problems by weak teachers denotes simple questions, retained in the first stage. The unsolved questions and the inconsistent CoT and PoT answers are advanced to the next stage for critical expert teaching. In the Critical Expert Teaching Stage, the expert critically reviews the problems that the weak model struggled to solve, providing a critique $r_j$ and supposed solution $\{A_j : (r_j, c_j, p_j) : j = 1, \ldots, N_2\}$. If the solution of the expert demonstrates external consistency, the problem and generated response are retained in this stage. Otherwise, the problem advances to the next stage. In this stage, we leverage a more advanced model, such as GPT-4o to ensure solution quality better than the weak teacher. In the Experts Consistency Voting Stage, multiple experts are engaged to provide solutions $\left\{ A_k : c_k^{(j)}, p_k^{(j)}; k = 1, \ldots, N_3; j = 1, \ldots, E \right\}$,

---

**Algorithm 1** `WISDOM` Dataset Construction

---

**Input:** Problems $\mathcal{Q}$, Meta info $\mathcal{M}$, Datasets $\mathcal{R}_1 = \emptyset$, $\mathcal{R}_2 = \emptyset$, $\mathcal{R}_3 = \emptyset$
**Output:** `WISDOM` Dataset $\mathcal{W} = \mathcal{R}_1 \cup \mathcal{R}_2 \cup \mathcal{R}_3$

1: **for** $i = 1$ to $Rounds$ **do**
2:  Initialize $\mathcal{S}_1, \mathcal{S}_2, \mathcal{S}_3 \leftarrow \emptyset$
3:  **for** $q_i \in \mathcal{Q}$ **do**
4:   Extract $m_i$; $\mathcal{M} \leftarrow \mathcal{M} \cup \{m_i\}$
5:   Generate $r_i = \text{CoT}(q_i) + \text{PoT}(q_i)$ via DeepSeek-Coder V2
6:   **if** $\text{CoT}(q_i) = \text{PoT}(q_i)$ **then**
7:    $\mathcal{S}_1 \leftarrow \mathcal{S}_1 \cup \{(q_i, m_i, \text{CoT}(q_i))\}$
8:   **else**
9:    $\mathcal{S}_2 \leftarrow \mathcal{S}_2 \cup \{(q_i, r_i)\}$
10:   **end if**
11:  **end for**
12:  **for** $(q_j, r_j) \in \mathcal{S}_2$ **do**
13:   Generate $r'_j$ using GPT-4o
14:   **if** $\text{CoT}(r'_j) = \text{PoT}(r'_j)$ **then**
15:    $\mathcal{S}_2 \leftarrow \mathcal{S}_2 \cup \{(q_j, m_j, \text{CoT}(r'_j))\}$
16:   **else**
17:    $\mathcal{S}_3 \leftarrow \mathcal{S}_3 \cup \{q_j\}$
18:   **end if**
19:  **end for**
20:  **for** $q_k \in \mathcal{S}_3$ **do**
21:   Generate $\{r_{k1}, ..., r_{kn}\}$; Vote for $r_k^*$
22:   **if** consistent $r_k^*$ **then**
23:    $\mathcal{S}_3 \leftarrow \mathcal{S}_3 \cup \{(q_k, m_k, \text{CoT}(r_k^*))\}$
24:   **end if**
25:  **end for**
26:  Cluster embeddings of $\mathcal{S}_3$ Meta info; Sample $m_1, m_2, m_3$
27:  Generate $q_{\text{new}}$ via GPT-4o with $m_1, m_2, m_3$ (40% mask); $\mathcal{Q} \leftarrow \mathcal{Q} \cup \{q_{\text{new}}\}$
28:  $\mathcal{R}_1 \leftarrow \mathcal{R}_1 \cup \mathcal{S}_1, \mathcal{R}_2 \leftarrow \mathcal{R}_2 \cup \mathcal{S}_2, \mathcal{R}_3 \leftarrow \mathcal{R}_3 \cup \mathcal{S}_3$
29: **end for**
30: **return** $\mathcal{W} = \mathcal{R}_1 \cup \mathcal{R}_2 \cup \mathcal{R}_3$

---

using a majority voting approach to address the remaining much more challenging questions. Meanwhile, we generate the meta-information related to these questions with the weak teacher, such as DeepSeek-Coder V2, along with the newly synthesized questions in the hard instruction evolving module with expert, *i.e.*, GPT-4o, serves as updated seed data for the next round. Throughout the Cyclic Iterative process, question difficulty is progressively increased, and we can obtain different level synthetic problems and generated response as the final dataset for fine-tuning.

## 2.3 COSTS AND RESPONSE EFFICIENCY BALANCING

For data synthesis, majority voting is an effective yet resource-intensive method to improve accuracy. Given that the data synthesis process may generate many trivial and vanilla problems (Tong et al., 2024), we apply majority voting specifically to evaluate and opt for the most difficult questions. Therefore, we first employ a weak but cost-effective teacher to solve a large number of easy problems. Subsequently, a strong but more resource-intensive expert is used for medium-difficulty questions, thereby optimizing resource utilization. To ensure the quality of generated responses, Rejection Sampling is an effective way. However, it relies heavily on the ground truth of seed data, making it less applicable to newly synthesized problems. We enforce unsupervised consistency to improve the quality of responses. With the weak teacher and inner consistency, we can filter out many simple, easily solvable problems early on, saving costs and improving efficiency. Although we also apply majority voting in the final stage to further increase response accuracy, by this point, the dataset has been significantly reduced in size and the difficulty of the problem has been greatly enhanced.

## 2.4 Question Diversity Enhanced via Meta Data Generation

Since data diversity plays a crucial role in instruction tuning, following existing work (Tang et al., 2024; Li et al., 2024a; Chan et al., 2024), we utilize the meta-information of questions to enhance the diversity of synthesized data. In mathematics, meta-information often refers to extracted skills or topics from questions, and synthesizing data can be generated based on this extra information. While existing methods (Tang et al., 2024; Huang et al., 2024) rely solely on skill or topic for question synthesis, we argue that focusing only on these aspects can be overly simplistic and limit the diversity of the synthesized questions. Instead, we extract a richer set of meta-information, including Skill, Topic, Knowledge Base, and Similar Problem, and combine multiple pieces of meta-information from various questions to maximize the diversity of the synthesized questions.

Specifically, we first extract meta-information from the questions (e.g., $q_3$) and obtain embeddings from the knowledge base in the meta-information using OpenAI's text-embedding-ada-002. We then apply k-means++ (Arthur & Vassilvitskii, 2007) clustering to group these embeddings. Furthermore, we randomly combine all of the aforementioned meta-information across different clusters and mask the Knowledge Base and Similar Problem with a 40% probability to prompt GPT-4o-0513 to generate new questions. This strategy not only prevents overfitting on the synthesized data but also significantly enhances the diversity of the generated questions.

---

**Example 2.1: Knowledge base contained Q-A Pair**

**Question:** A biologist is tracking the growth of a bacterial culture that doubles in size every 3 hours. If the initial size of the culture is 500 bacteria, how many bacteria will be present after 15 hours?
**Response:**

`<knowledge_base_start>`

Key Definitions and Explanations:. . . - Exponential Growth: . . .

`</knowledge_base_end>`

`<solution_start>`

Step-by-Step Solution:1. Understand the Problem: . . . The final answer is: boxed{16000}

`</solution_end>`

---

## 2.5 Knowledge-based curriculum learning

To further enhance the quality of synthesized data and improve the model's reasoning abilities, the current mainstream approach is to adopt the Chain of Thought (CoT) method, which transforms simple seed data responses into structured, step-by-step reasoning processes and teaches the model to think before solving. Inspired by the principles of curriculum learning, it is advantageous to recall key knowledge points relevant to the problem before answering a question. Thus, we integrate the knowledge base from the meta-information, which includes Key Definitions and Explanations, Relevant Formulas and Equations, Common Pitfalls and Misconceptions, and Additional Theoretical Insights, into the preamble of the CoT response. This step helps the model reinforce critical knowledge points before proceeding to solve a given question. A simple illustrative example can be found in Example 2.1, and more detailed examples can be found in Appendix D.

## 3 EXPERIMENTS

### 3.1 Evaluation datasets

To evaluate curriculum learning in recent benchmarks, we select a diverse and challenging set of in-domain and out-of-domain benchmarks for evaluation. In-domain datasets include GSM8K (Cobbe et al., 2021) and MATH (Hendrycks et al., 2021) and out-of-domain datasets contain College MATH (Tang et al. (2024)), OlympiadBench-Math (He et al. (2024)), TabMWP (Lu et al. (2024a)), TheromQA (Chen et al. (2023)), AMC2023 and AIME2024. These selected datasets are to compre-

hensively assess the model's reasoning ability to solve mathematically challenging problems across various dimensions. More details of datasets can be found in Appendix A.3.

## 3.2 BASELINES

We select several representative state-of-the-art closed-source models and open-source models for comparsion, including GPT-4o-0513, GPT-4-1106-preview, Claude-3-Opus, Llama3-series (AI@Meta (2024)), DeepSeek-Math-Instruct(Shao et al. (2024))(denote DSMath), DeepSeek-Coder-V2 (Zhu et al. (2024)), Qwen-2-72B-instruct (Yang et al. (2024)), MetaMath (Yu et al. (2023)),MathScale (Tang et al., 2024), MAmmoTH2 (Yue et al. (2024b)), KPMath-Plus (Huang et al. (2024)), DART-Math (Tong et al. (2024)), NuminaMath (Li et al. (2024c)), Mathstral(Team, 2024). Comparison of `WISDOM` and others can be found in Table 1.

Table 1: Comparison of `WISDOM` and other mathematical synthetic methods.

| Method | Cost Efficiency | GT-Free | w/o External Data | Instruction Diversity Evolution | Instruction Difficulty Evolution |
|---|---|---|---|---|---|
| MetaMath | ✗ | ✗ | ✓ | ✓ | ✗ |
| MathScale | ✗ | ✓ | ✓ | ✓ | ✗ |
| MAmmoTH2 | ✗ | ✓ | ✗ | ✓ | ✗ |
| KPMath-Plus | ✗ | ✓ | ✗ | ✓ | ✗ |
| DART-Math | ✓ | ✗ | ✓ | ✗ | ✗ |
| NuminaMath | ✗ | ✗ | ✗ | ✓ | ✗ |
| `WISDOM` (ours) | ✓ | ✓ | ✓ | ✓ | ✓ |

Table 2: Main results on in-domain benchmarks, GSM8K and MATH, and out-of-domain benchmarks, including College MATH, Olympiad, TabMWP, TheoremQA, AMC2023, and AIME2024. We select the current well-performing LLMs to evaluate their test accuracy on these benchmarks. Since KPMath-Plus is not open-sourced, the results are quoted from the corresponding paper.

| Method | Base | GSM8K | MATH | College[†] | Olympiad | TabMWP | TheoremQA | AMC2023 | AIME2024 |
|---|---|---|---|---|---|---|---|---|---|
| GPT-4o-0513 | – | 95.8 | 76.6 | – | – | – | – | – | 2/30 |
| GPT-4-1106-preview | – | 91.4 | 64.3 | – | – | – | – | – | 1/30 |
| Claude-3-Opus | – | 95.0 | 60.1 | – | – | – | – | – | 2/30 |
| DeepSeek Coder V2 | – | 94.9 | 75.7 | – | – | – | – | – | 4/30 |
| Mathstral | | **83.3** | 54.3 | 36.7 | **22.4** | **82.8** | 26.3 | 12/40 | **1**/30 |
| KPMath-Plus | | 82.1 | 46.8 | – | – | 66.4 | – | – | – |
| DART-Math | Mistral-7B | 81.3 | 45.0 | 28.3 | 14.5 | 65.8 | 20.5 | 7/40 | 0/30 |
| MAmmoTH2 | | 67.4 | 34.2 | 31.0 | 9.8 | 26.8 | 26.7 | 6/40 | 1/30 |
| MathScale | | 58.5 | 33.2 | 22.0 | 7.8 | 73.3 | 18.1 | 6/40 | 1/30 |
| `WISDOM` | | 80.0 | 56.4 | 41.6 | 21.9 | 72.3 | 27.6 | **15**/40 | 1/30 |
| Llama3-instruct | | 78.2 | 27.2 | 22.8 | 5.6 | 75.3 | 18.9 | 5/40 | 0/30 |
| MetaMath | | 80.5 | 32.6 | 19.3 | 6.7 | 54.1 | 13.3 | 6/40 | 0/30 |
| DART-Math | Llama3-8B | 81.8 | 46.9 | 28.4 | 15.9 | 66.3 | 20.5 | 8/40 | **1**/30 |
| MAmmoTH2 | | 69.6 | 33.4 | 32.3 | 8.1 | 43.8 | **29.7** | 7/40 | 0/30 |
| MathScale | | 70.8 | 34.6 | 22.5 | 9.0 | 74.3 | 18.9 | 2/40 | 1/30 |
| `WISDOM` | | **83.2** | **59.7** | **42.2** | **25.6** | **83.0** | 28.6 | **17**/40 | 1/30 |
| DSMath-instruct | | 82.0 | 46.3 | 38.1 | 13.6 | 76.7 | 31.9 | 7/40 | 1/30 |
| MetaMath | | 76.5 | 37.2 | 27.3 | 10.7 | 67.1 | 13.9 | 10/40 | 0/30 |
| KPMath-Plus | | 83.9 | 48.8 | – | – | 78.7 | – | – | – |
| DART-Math | DSMath-7B | **87.5** | 53.9 | 40.7 | 20.0 | 82.9 | 31.5 | 8/40 | 0/30 |
| NuminaMath | | 77.1 | 53.7 | 32.4 | 24.0 | 77.7 | 29.4 | **12**/40 | 1/30 |
| MathScale | | 62.7 | 33.4 | 23.0 | 8.1 | 71.3 | 24.5 | 4/40 | 0/30 |
| `WISDOM` | | 83.3 | **62.4** | **45.0** | **28.9** | **85.7** | **34.9** | 11/40 | **2**/30 |
| Llama3-instruct | Llama3-70B | 93.1 | 50.4 | 40.3 | 17.6 | 89.9 | 34.1 | 8/40 | 2/30 |
| Qwen2-instruct | Qwen2-72B | 93.6 | 69.3 | 46.8 | 35.3 | 92.4 | 42.0 | 17/40 | **4**/30 |
| DART-Math | Llama3-70B | 89.8 | 55.7 | 37.9 | 21.0 | 80.9 | 28.2 | 13/40 | 1/30 |
| KPMath-Plus | Qwen1.5-72B | 87.0 | 58.3 | – | – | 76.7 | – | – | – |
| MetaMath | Llama3-70B | 88.0 | 44.9 | 31.9 | 11.6 | – | 21.9 | – | – |
| NuminaMath | Qwen2-72B | 91.5 | 66.9 | 42.1 | 33.6 | 86.7 | 29.0 | 13/40 | **4**/30 |
| `WISDOM` | Llama3-70B | 94.1 | 68.2 | 43.4 | 34.4 | 91.8 | 41.4 | 22/40 | 3/30 |
| `WISDOM` | Qwen2-72B | **94.2** | **76.1** | **47.6** | **39.1** | **94.5** | **45.4** | **23**/40 | 2/30 |

[†] In short of College MATH.

## 3.3 MAIN RESULTS

As shown in Table 2, we present the performance of `WISDOM` on in-domain and out-of-domain datasets, which demonstrates it achieves strong results across all datasets, particularly excelling on

challenging ones, such as AIME2024, AMC2023. Notably, `WISDOM` has set a new SOTA performance on the MATH dataset. Based on the same small model DSMath-7B, our method reaches a significant milestone on the in-domain MATH dataset, surpassing the 60% threshold for the first time with a score of 62.4%, compared to the previous SOTA DART-Math achieving 53.9%. In addition, even for a weaker mathematical foundation model, we can achieve remarkable improvements. While based on the same base model mistral-7B, Mathstral achieved 54.3%, `WISDOM` reaches 56.4%, marking a 2.1% improvement. Meanwhile, based on Llama3-8B, previous SOTA DART-Math achieves 46.9%, `WISDOM` reaches 59.7%, marking a 12.8% improvement.

As shown in Table 2, on the different out-of-domain datasets, our method exhibits stronger generalization capabilities, enhancing the ability of smaller models to tackle challenging mathematical problems. Specifically, based on Llama3-8B, DSMath-7B, Qwen2-72B, and Llama3-70B, our method `WISDOM` achieves new SOTA on College MATH, Olympiad, TabMWP. For more challenging tasks, such as TheoremQA, AMC2023 and AIME 2024, our `WISDOM` model demonstrates outstanding performance, even when built on smaller model foundations. For instance, based on Llama3-8B, we successfully solve 17 out of 40 questions whereas the current SOTA method on the same model base only solved 8, which represents a relative 112.5%(17/8) improvement, matching the performance of Qwen2-Instruct. Remarkably, on the AIME 2024 dataset, our method performs on par with GPT-4o, suggesting the potential of smaller models in solving complex mathematical problems with easy-to-hard curriculum learning. Even when using only synthetic data, the model shows excellent performance on both in-domain and out-of-domain datasets, notably surpassing GPT-4o on the AIME 2024 dataset. Furthermore, we conduct experiments on two larger models (Llama3-70B and Qwen2-72B) to explore the upper limits of model performance. The results indicate that our approach remains effective at this scale. For example, by implementing easy-to-hard curriculum learning on Llama3-70b, we improve the performance on the AMC2023 from 8/40 to 22/40, surpassing the performance of Qwen2-Instruct-72B.

Overall, the main results above clearly demonstrate that our proposed easy-to-hard curriculum learning based synthesis method (`WISDOM`) is highly effective in improving performance across both small and large models.

### 3.4 THE IMPACT OF SCALING STAGE

As illustrated in Figure 4, we conduct several experiments on MATH to investigate the impact of scaling stages in our data synthesis method. Initially, the DeepSeek-Math-7B-base and LLaMA3-8B-base achieve accuracies of only 36.2% and 20.5%, respectively. However, fine-tuning with seed data improved their performance to 52.22% and 36.68%, resulting in gains of 16.02 and 16.18 percentage points.

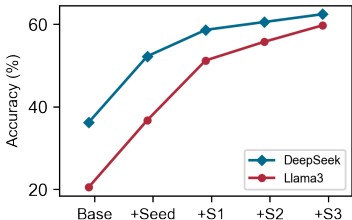

Figure 4: The accuracy of MATH in relation to the scaling effects across different stages.

With the data synthesized from Weak Teacher Guiding, the models' capabilities improve further to 58.56% (+6.34%) and 54.04% (+17.36%), respectively. After additional fine-tuning in Critical Expert Teaching, which primarily focused on medium-difficulty questions, performance increases to 60.52% (+1.96%) and 57.44% (+3.40%). Ultimately, in Experts Consistency Voting, the model achieves accuracy of 62.44% (+1.92%) and 59.72% (+2.28%). These experimental results clearly demonstrate the effectiveness of our data synthesis method in enhancing complex mathematical reasoning abilities at each phase for small-scale base models. More results on out-of-domain datasets can be found in Appendix C.

### 3.5 THE IMPACT OF DIFFERENT STAGES

To validate the impact of each stage in the progressive curriculum synthesis process on enhancing mathematical reasoning, we randomly sample 200k data points from each stage and perform supervised fine-tuning (SFT) in conjunction with the initial seed data, GSM8K and MATH training sets. As shown in Table 3, after fine-tuning, we observe that the Experts Consistency Voting stage led to a significant accuracy improvement on all challenging problems compared to the other stages, achiev-

Table 3: Ablation results on Llama3-8B fine-tuned across different stages of WISDOM.

| Seed | S1 | S2 | S3 | GSM8K | MATH | TheromQA | CollegeMATH | Olympiad | AIME 2024 | AMC 2023 | TabMWP |
|------|----|----|----|-------|------|----------|-------------|----------|-----------|----------|--------|
| ✓ | ✓ | | | **80.4** | 42.8 | 25.1 | 25.6 | 15.3 | 1/30 | 4/40 | 76.2 |
| ✓ | | ✓ | | 78.2 | 42.7 | 23.4 | 26.2 | 16.5 | 1/30 | **7**/40 | 76.4 |
| ✓ | | | ✓ | 80.3 | **47.1** | **25.1** | **30.8** | **18.1** | **1**/30 | 5/40 | **78.3** |

ing 47.14%, 25.13%, 30.76%, 18.10%, 78.3% on MATH, TheromQA, CollegeMATH, Olympiad and TabMWP respectively. However, the Critical Expert Teaching stage does not show substantial improvement over the Weak Teacher Guiding. Actually, only a small portion of answers were modified during the Critical Expert Teaching stage. For most answers, CoT responses from the Weak Teacher Guiding were retained, some were converted to PoT answers, and a few were completely revised. This finding also explains the observed improvements on challenging datasets. Overall, as the stages progress, the difficulty of the curriculum learning process steadily increases, contributing to performance enhancement.

### 3.6 INFLUENCE OF KNOWLEDGE BASE ON ANSWERS

Table 4: Comparison of performance on different benchmarks with and without the inclusion of the knowledge base. To validate the generalizability of the approach, experiments were conducted on both Llama3 and DeepSeek-Math.

| Model | GSM8K | MATH | CollegeMATH | Olympiad | TabMWP | TheromQA | AIME 2024 | AMC 2023 |
|-------|-------|------|-------------|----------|--------|----------|-----------|----------|
| WISDOM (Llama3-8B-base) | 83.1 | **59.7** | **42.2** | 25.6 | **83.0** | 28.6 | 1/30 | **17**/40 |
| w/o knowledge base | **83.9** | 59.6 | 41.7 | 25.3 | 82.1 | **32.5** | 1/30 | 9/40 |
| WISDOM (DSMath-7B-base) | 83.3 | **62.4** | **45.0** | 28.9 | **85.7** | 34.9 | 2/30 | 11/40 |
| w/o knowledge base | 79.5 | 58.6 | 42.4 | 23.7 | 85.3 | 31.8 | 1/30 | **12**/40 |

Table 5: Comparison of performance on MATH with and without knowledge base using sample data in Weak Teacher Guiding stage.

| data | knowledge base | Model | Math |
|------|----------------|-------|------|
| Seed+S1(100k) | ✗ | Llama3-8B | 39.0 |
| Seed+S1(100k) | ✓ | Llama3-8B | 43.1(+4.1%) |
| Seed+S1(200k) | ✗ | Llama3-8B | 43.2 |
| Seed+S1(200k) | ✓ | Llama3-8B | 45.7(+2.5%) |
| Seed+S1(400k) | ✗ | Llama3-8B | 46.8 |
| Seed+S1(400k) | ✓ | Llama3-8B | 49.0(+2.2%) |

Table 6: Accuracy w/ and w/o consistency.

| | Wrong Number | Right Number | Acc. (%) |
|---|---|---|---|
| w/o consistency | 92,696 | 14,304 | 13.4 |
| w/ consistency | 79,966 | 27,034 | 25.3(+11.9) |

To investigate the impact of knowledge base on learning within the curriculum learning process, we conduct experiments to explore its effects at different scales. As shown in Table 5, we randomly select data from the Weak Teacher Guiding stage and observe that as the data scale increases, the results with knowledge base consistently outperform those without it, with a minimum improvement of 2% on the MATH. However, we also notice that the rate of improvement decreases as the data volume increases, prompting us to conduct a full-scale experiment. As shown in Table 4, we conduct experiments on Llama3 and DeepSeek-Math, removing the knowledge base from the process. We find that the difference in performance between the presence and absence of the knowledge base is less pronounced when dealing with smaller datasets. However, as the data scale increases, the contribution of the knowledge base to performance becomes more evident, though the differences are not dramatic in most metrics. This may because the model can infer some of the knowledge from the answers themselves, similar to how one might deduce knowledge points from the context of an answer. We also observe that DeepSeek-Math is more efficient at learning from knowledge points compared to Llama 3. We hypothesize that this is because the meta-information is generated by DeepSeek-Coder V2, which may include pre-training data relevant to DeepSeek-Math, thereby better activating the knowledge learned during its pre-training. The results indicate that scaling the knowledge base to a million-level scale still yields improvements, underscoring the importance of knowledge point learning in the curriculum learning process.

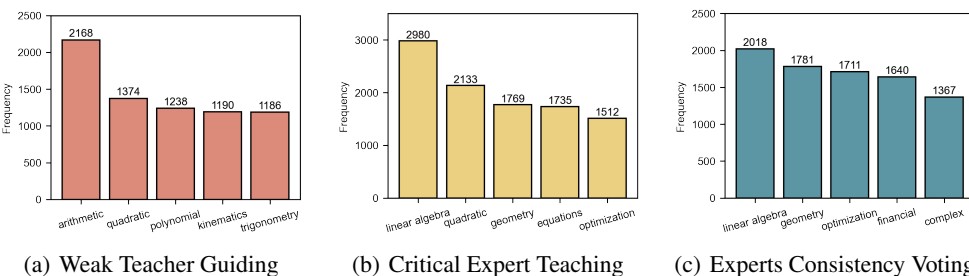

(a) Weak Teacher Guiding     (b) Critical Expert Teaching     (c) Experts Consistency Voting

Figure 5: The top 5 topics and their corresponding frequencies after clustering in the three stages.

### 3.7 EFFECT OF ANSWER CONSISTENCY

We first sample 100k synthetic data points, each containing two different types of responses to the same set of questions: one consistent and one inconsistent. To investigate their accuracy, we use GPT-4o to generate reference answers for the 100k synthetic data points and compare the generated responses with the reference answers to measure consistency. Given that GPT-4o is currently among the most advanced models in terms of mathematical capabilities, we can reasonably assume that consistency rates closely reflect accuracy. As shown in Table 6, consistent responses improve accuracy by 11.9%. To further explore the impact of response consistency on model training, we conduct an ablation study by replacing all consistent data in Weak Teacher Guiding stage with data lacking consistency. As illustrated in Table 8 in Appendix E, it is evident that the absence of consistency resulted in a significant decline in performance across all datasets. Response consistency not only enhances accuracy and helps increase the difficulty of synthesized instructions but also contributes to improving the model's mathematical reasoning abilities during training.

### 3.8 ANALYSIS

#### 3.8.1 DATASETS EMBEDDING TOPICS

We conduct an in-depth analysis of the topics included in each stage of the problems, providing direction for future synthesis of mathematical reasoning datasets. Specifically, we sample 100k data points from each of the three stages, convert the knowledge base into embeddings using OpenAI's text-embedding-ada-002, and apply K-means++ (Arthur & Vassilvitskii, 2007) for clustering, resulting in 200 categories. For each category, we extract the central terms and identify the top five topics along with their frequencies. As shown in Figure 5, In the initial stage, the data predominantly features simple and clear topics, such as arithmetic and kinematics. As the difficulty increases, more complex subjects emerge, including linear algebra, optimization, and complex numbers. In the final stage, financial-related problems appear, possibly due to the model's difficulty in handling decimal precision in financial interest calculations, where the reasoning path is correct, but the answers are wrong. Therefore, future synthesis of mathematical reasoning problems should aim for more refined generation and optimization, particularly targeting similar topics while balancing computational precision and cognitive complexity.

#### 3.8.2 COST SAVING

Curriculum learning not only facilitates a gradual learning process but also enables the strategic allocation of more resources to difficult problems. This approach can significantly reduce costs compared to the traditional majority voting method. We analyze the number of tokens used and the overall expenditure, as illustrated in Figure 6. By calculating the average token consumption for inputs and outputs within the sampled dataset, and applying an exchange rate of 1:7 between USD and CNY, we determine the overall cost of the dataset based on the API pricing for DeepSeek and GPT-4o-0513. Our analysis shows that our approach is 2.82 times more cost-effective compared to majority voting, leading to a total savings of over 20,000 US dollars in overall expenditure. Our method resulted in a substantial reduction in costs while maintaining strong outcomes, demonstrating the scalability and cost-effectiveness of our approach.

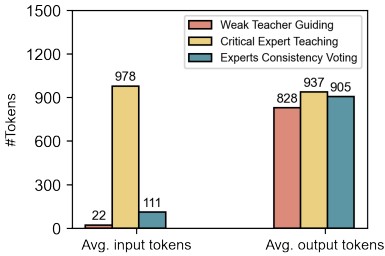 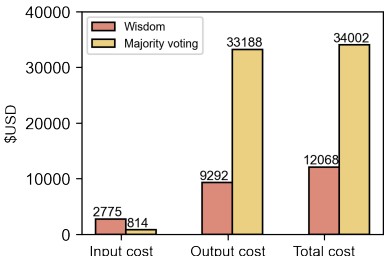

Figure 6: The left figure illustrates the average number of input and output tokens at different stages, while the right figure presents the monetary cost associated with input and output tokens, as well as the total expenditure.

## 4 RELATED WORK

**Instruction Tuning**. Instruction fine-tuning has been widely validated as an effective method for enhancing model capabilities. Previous work (Kang et al. (2024); Chen et al. (2024); Xia et al. (2024); Li et al. (2024d)) has primarily focused on improving model performance through the careful selection of high-quality data. However, recent literature (Tang et al. (2024); Chung et al. (2022); Yue et al. (2024b); Tang et al. (2024)) suggests that increasing the volume of data can also contribute to performance gains. Unlike efforts (Zhang et al. (2024); Muennighoff et al. (2023)) that aim to broadly enhance general model capabilities, our instruction fine-tuning is specifically designed to improve complex mathematical reasoning skills.

**Mathematical Reasoning**. To enhance answer accuracy on challenging benchmarks, recent research (Gou et al. (2024); Zhou et al. (2024); Wang et al. (2024b)) has increasingly focused on leveraging external tools to improve large language models' (LLMs) ability to solve mathematical problems and achieve higher scores. Most approaches utilize Program of Thought (PoT) methods, employing code interpreters to compute the final result (numerous methods cited here). Additionally, some methods (Wang et al. (2023a); Aggarwal et al. (2023); Wang et al. (2024a); Shao et al. (2024)) adopt self-consistency techniques, ensembling multiple outputs to achieve better results. In contrast, we concentrate on improving the model's intrinsic reasoning capabilities, relying solely on Chain of Thought (CoT) approaches to develop the model's inherent mathematical reasoning skills.

**Distillation**. Training smaller student models using synthetic data generated by more powerful teacher models has been widely validated as an effective approach (Xu et al. (2024); Li et al. (2024f); Wang et al. (2024c); Li et al. (2024b)) . In the domain of mathematical reasoning, this effectiveness has also been demonstrated (Yu et al. (2024); Yue et al. (2024a); Tang et al. (2024); Li et al. (2024a); Azerbayev et al. (2024)). However, these methods still exhibit a significant gap in performance when tackling challenging mathematical problems compared to closed-source models. Our work addresses this gap by employing curriculum learning, which not only synthesizes diverse and challenging responses but also gradually generates problems with greater diversity and complexity, thereby significantly narrowing the performance gap with closed-source models.

## 5 CONCLUSION

In this work, we propose WISDOM as a data-synthesis framework to enhance the mathematical reasoning abilities of LLMs. The key insight behind WISDOM is the use of progressive curriculum synthesis, which iteratively evolves the difficulty of questions, and generates high-quality responses in unsupervised ways based on response consistency of weak and strong models without relying on ground truths. Compared to traditional majority voting, WISDOM is 2.82x more cost-effective. To validate the effectiveness of the synthesized datasets, we fine-tune a series of open-sourced LLMs ranging from 7B to 72B parameters. Experimental results show that the fine-tuned LLMs achieve significant improvements over the base models, highlighting the generalization capabilities of WISDOM. Our empirical findings also offer valuable insights into tackling challenging problems, paving the way for future complex reasoning across various fields.

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

# A EXPERIMENTAL SETUP

## A.1 TRAINING SETUP

We employ Llama factory (Zheng et al. (2024)) for fine-tuning the entire suite of models and utilized sequence packing Krell et al. (2021) to accelerate the training process. To accommodate the large model sizes during training, we leveraged DeepSpeed with ZeRO (Rajbhandari et al. (2020)) Stage 3. For data preprocessing, we applied the Alpaca prompt format (Taori et al. (2023)).

Table 7: The composition and sources of data for the proposed WISDOM dataset.

| Dataset | Pairs | Dataset Source |
|---|---|---|
| **Weak Teacher Guiding** | 640K | DeepSeek Coder V2 |
| **Critical Expert Teaching** | 527K | GPT-4o-0513 |
| **Experts Consistency Voting** | 300K | GPT-4o-0513 |

The training was conducted using 88 NVIDIA A800 GPUs, with a configuration of batch size 1, gradient accumulation (Lin et al. (2018)) of 2, sequence length of 8192, and bf16 (Kalamkar et al. (2019)) precision. We optimized the models with the AdamW (Loshchilov & Hutter (2019)) optimizer, setting a learning rate warmup using a cosine schedule with a warmup ratio of 0.03, and trained each model for 3 epochs. The learning rates were adjusted slightly for different models: Mistral 7B at 1e-5, DeepSeekMath-7B at 5e-5, Llama3-8B at 4e-5, and both Llama3-70B and Qwen2-72B at 2e-5.

All final results for the models were obtained using the full dataset. The specific composition of the dataset can be found in Table 7.

## A.2 DATA CONTAMINATION

To mitigate the risk of data contamination, we applied a 10-gram hash deduplication method (Liu et al. (2019)) to the questions in both our in-domain and out-of-domain benchmarks, with a condition that the ratio of the longest common sequence must exceed 0.6 (Yang et al. (2024)), Any detected duplicates were removed. According to our statistics, after deduplication, the remaining samples were as follows: from 641,514 to 640,987 in Weak Teacher Guiding, from 527,658 to 527,537 in Critical Expert Teaching, and from 298,190 to 298,118 in Experts Consistency Voting Stage. All experiments were conducted on these deduplicated datasets to prevent potential data contamination.

## A.3 DETAIL DATASETS

The following provides a detailed description of the composition of the evaluation set.

- GSM8K: The test dataset consists of 1,319 high-quality grade school mathematics problems, primarily to evaluate fundamental logical reasoning and applied mathematical abilities.

- MATH: 5,000 curated high school competition-level test problems, including diverse dimensions like Prealgebra, Algebra, Number Theory, Counting and Probability, Geometry, Intermediate Algebra, and Precalculus.

- College MATH: A total of 2,818 college-level mathematics problems were extracted from nine textbooks covering seven domains. Mathematical reasoning ability can be assessed from multiple skill perspectives, such as analytical thinking, logical reasoning, and quantitative analysis.

- OlympiadBench-Math: The text-only English subset of Olympiad-Bench, consisting of 675 Olympiad-level mathematical problems, is designed to evaluate complex and advanced mathematical reasoning abilities.

- TabMWP: A large-scale dataset for math word problems in tabular contexts.The test dataset includes 1,000 questions with tabular contexts, allowing for a comprehensive evaluation of mathematical reasoning within the context of tables.

- AMC-AIME: The datasets are designed to select students who will represent the United States at the International Mathematics Olympiad (IMO). The datasets include 30 competition-level problems from the AIME2024 and 40 from the AMC2023, covering a broad spectrum of problem-

solving skills such as arithmetic, algebra, combinatorics, geometry, number theory, and probability.

## B PROMPTS

### B.1 META-INFORMATION GENERATION PROMPT

---

**Example B.1: Meta-information Generation Prompt**

As a math mentor, you are dedicated to helping your students understand and master key mathematical concepts and problem-solving techniques. Your goal is to provide clear, concise guidance and support. When guiding, you must not give out the answer to the original problem.

```
## Skill Label
```
Consider the following mathematical question. Label this question with a specific mathematical skill required to solve it. The skill name should:

- Be in lowercase letters only.
- Be very descriptive.
- Use multiple words joined by an underscore if necessary.
- Enclose the content within `<skill>` and `</skill>` tags.

```
## Topic Label
```
Consider the following mathematical question. Label this question with the specific mathematical topic it belongs to. The topic name should:

- Be in lowercase letters only.
- Be specific and descriptive.
- Use underscores to join multiple words if necessary.
- Enclose the content within `<topic>` and `</topic>` tags.

```
## Knowledge Base
```
Provide comprehensive information necessary for understanding the mathematical concepts related to the given problem, without including step-by-step procedures or any information that could directly solve the problem. Include the following:

- Key definitions and explanations.
- General relevant formulas and equations (without applying them to the specific problem).
- Common pitfalls and misconceptions.
- Additional theoretical insights related to the topic.
- Do not include any visual or diagram-related knowledge.
- Enclose the content within `<knowledge_base>` and `</knowledge_base>` tags.

```
## Similar Problem Types
```
Provide up to two examples and solutions of similar problem types to help students recognize patterns and apply similar problem-solving methods. For each example:

- State the problem.
- Provide a detailed solution.
- Highlight the similarities to the original question.
- Explain how the solution method can be applied to the original question.
- Do not include any visual or diagram-related knowledge.
- Enclose the content within `<similar_problems>` and `</similar_problems>` tags.

```
## Question
```
{question}

---

## B.2   QUESTION GENERATION PROMPT

---

**Example B.2: Question Generation Prompt**

**Role**: You are a creative math professor.

**Objective**: Help senior students learn the following math key skills and topics, create high quality math word problems to help students learn math.

**Task**:

1. Using the listed key skills and topics as guidelines, construct multiple, original math problems. Each problem should be written on a new line.

2. Ensure each problem has a fixed and unique answer.

3. Increase the difficulty of the problems by incorporating foundational knowledge and common pitfalls.

4. Problems can be generated using a single topic and skill or by combining multiple topics and skills for higher quality questions.

5. Reference the given problems and maximize the combination of topics and skills to rewrite and deepen the difficulty, ensuring no conflicts between topics and skills.

6. Based on the example problem, imagine what specific field of study would delve into such data and generate the problem as if it were created by someone in that field.

7. Ensure that the generated questions are solvable and not inherently unsolvable.

8. Each generated question must be a single question without any sub-questions.

9. Ensure that the generated questions are as quantitative as possible, focusing on problems that can have numerical solutions.

**Instructions**:

- Write each new math problem on a new line.
- Use `<question>` to indicate the beginning of the question.
- Use `</question>` to indicate the end of the question.

**Topics and skills**:

- Skills: {skills}
- Topics: {topics}

**Knowledge_base**:

> {knowledge_base}

**Easy example problems**:

> {problems}

**Expanded Requirements**:

1. Generate up to eight questions per response, with each question on a new line.

2. Each generated question must have a fixed and unique answer.

3. Increase the difficulty of the questions using foundational knowledge and common pitfalls.

4. Use a single topic and skill or combine multiple topics and skills to create higher quality questions.

5. Reference the given questions and maximize the combination of topics and skills to rewrite and deepen the difficulty, ensuring no conflicts between topics and skills.

6. Each generated question must be a single question without any sub-questions.

7. Ensure that the generated questions are as quantitative as possible, focusing on problems that can have numerical solutions.

---

## B.3 CRITICAL EXPERT TEACHING PROMPT

---

**Example B.3: Critical Expert Teaching Prompt**

**Task**: Solve a Complex Math Problem with Step-by-Step Explanation and Python Code

**Instructions:**

1. Read the math problem carefully.

2. Compare the reference solution and code output, identify discrepancies, and analyze the reasons for these discrepancies in the section between `<reference_solution_analysis>` and `</reference_solution_analysis>`.

3. Think through the correct solution step-by-step.

4. At the end of your step-by-step solution, provide the final answer in the format: `boxed{final_result}`.

5. After presenting the final answer, write a Python code that demonstrates the solution process.

6. **Ensure the final output of the code is stored in a variable named `result`.**

7. Provide your final solution in the section between `<solution>` and `</solution>`.

### BACKGROUND

I am working on a complex math problem and need your assistance. I have an incorrect reference solution and a result obtained from running a code. Please help me understand the discrepancies and find the correct solution.

### REFERENCE SOLUTION

{reference_solution}

### CODE OUTPUT

{code_output}

### PROBLEM

{question}

### ANALYSIS

`<reference_solution_analysis>`
Please analyze the discrepancies between the reference solution and the code output here.
`</reference_solution_analysis>`

### SOLUTION

`<solution>`
Please provide a step-by-step solution here, including the final answer in the format: `boxed{final_result}`.
After presenting the final answer, write a Python code that demonstrates the solution process.
Make sure the final answer in the format `boxed{final_result}` is presented before the Python code.
Ensure the final output of the code is stored in a variable named `result`.
`</solution>`

---

## B.4 WEAK TEACHER GUIDING PROMPT

---

**Example B.4: Weak Teacher Guiding Prompt**

You are a math professor and good at writing python code.
**Task**: Solve a Complex Math Problem with Step-by-Step Explanation and Python Code
**Instructions:**

1. Read the math problem carefully.

2. Think through the solution step-by-step.

3. At the end of your step-by-step solution, provide the final answer in the format: `boxed{final_result}`.

4. After presenting the final answer, write a Python code that demonstrates the solution process.

5. The generated code should not include any visualization libraries such as matplotlib.

**Problem**: {question}

---

## B.5 EXPERTS CONSISTENCY VOTING PROMPT

---

**Example B.5: Experts Consistency Voting Prompt**

**Problem:**
{question}
Answer the problem step by step, ensuring that each step is clearly explained and logically connected to the next. While the solution process should be detailed and thorough, ensure that the final answer is formatted as `\boxed{final_result}`.

---

## C  DETAILED SCALING ACCURACY

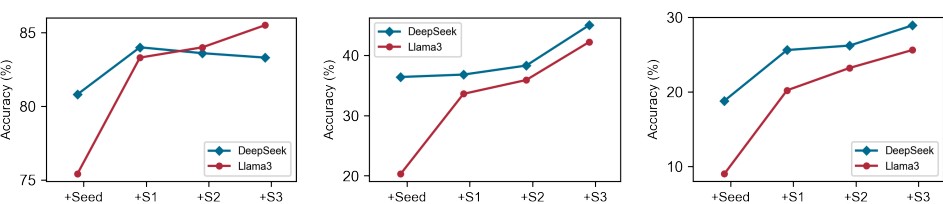

Figure 7: The accuracy results from left to right are for GSM8K, College Math, and Olympiad, respectively.

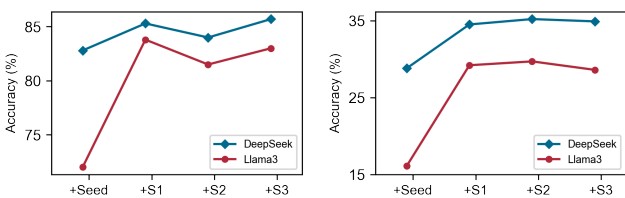

Figure 8: The accuracy results from left to right are for TabMWP, TheoremQA, respectively.

# D    DETAILED SAMPLES

---

**Example D.1: Weak Teacher Guiding Sample**

**Question:** In a certain chemical reaction, the ratio of reactant A to reactant B must be 3:7. If you have 21 grams of reactant B, how many grams of reactant A do you need?

**Response:**

`<knowledge_base_start>`

`### Key Definitions and Explanations`
- **Ratio**: A ratio compares two quantities, often expressed as 3:7, meaning that for every 3 parts of one substance, there are 7 parts of another.
- **Proportion**: A proportion is an equation that shows two ratios are equal. Proportions help solve for unknown values when part of the ratio is given.
`### General Relevant Formulas and Equations`
- **Proportion Formula**: If two ratios are equal, they can be set up as a proportion: $\frac{a}{b} = \frac{c}{d}$, where $a$ and $b$ are the known quantities, and $c$ or $d$ is the unknown to be solved.
`### Common Pitfalls and Misconceptions`
- **Reversing the Ratio**: Be sure to follow the correct order in ratios. For example, the ratio A:B is not the same as B:A.
- **Incorrect Proportional Set-Up**: Misinterpreting how to set up the proportion can lead to incorrect solutions. It's important to align the ratio and the actual quantities properly.
`### Additional Theoretical Insights`
- **Application of Ratios in Chemistry**: Ratios and proportions are critical in chemistry when mixing reactants in fixed proportions to ensure reactions occur as intended.
- **Scaling Proportions**: In practical scenarios, such as scaling up chemical reactions, understanding proportions is key to calculating the correct amounts of reactants.

`</knowledge_base_end>`

`<solution_start>`

`### Step-by-Step Solution:`
1. **Understand the ratio**: The ratio of reactant A to reactant B is given as 3:7. This means for every 3 parts of reactant A, there are 7 parts of reactant B.
2. **Set up the proportion**: Let $x$ be the amount of reactant A needed. Using the ratio, we set up the proportion:

$$\frac{x}{21} = \frac{3}{7}$$

where 21 is the given amount of reactant B.
3. **Solve for $x$**: Cross-multiply to solve for $x$:

$$7x = 3 \times 21$$

4. **Calculate the right-hand side**: Perform the multiplication:

$$3 \times 21 = 63$$

5. **Solve for $x$**: Divide both sides by 7:

$$7x = 63$$

$$x = \frac{63}{7} = 9$$

`### Final Answer:`

$$\boxed{9}$$

Thus, you need 9 grams of reactant A.

`</solution_end>`

---

**Example D.2: Critical Expert Teaching Sample**

**Question:** Find the values of $x$ and $y$ that satisfy the system of equations $x^2 + y^2 = 10$ and $x^2 - y^2 = 6$.

**Response:**

```
<knowledge_base_start>

### Key Definitions and Explanations
```
- **System of Equations**: A set of equations with the same variables, solved simultaneously.
- **Substitution and Elimination Methods**: Substitution solves one equation for one variable and substitutes it into the other equation. Elimination adds or subtracts equations to eliminate one variable.
- **Algebraic Operations**: Basic operations (addition, subtraction, multiplication, division) used to manipulate and solve equations.
```
### General Relevant Formulas and Equations
```
- For two equations like $x^2 + y^2 = 10$ and $x^2 - y^2 = 6$, adding or subtracting them simplifies the system and isolates a variable.
```
### Common Pitfalls and Misconceptions
```
- **Misinterpreting Operations**: Errors in tracking signs during addition/subtraction.
- **Square Root Ambiguities**: Both positive and negative roots must be considered.
```
### Additional Theoretical Insights
```
- **Graphical Interpretation**: Each equation represents a curve, and solutions are where these curves intersect.

```
</knowledge_base_end>

<solution_start>
```

Let's solve the given system of equations step-by-step:

1. **Understand the Problem**: We are given a system of two equations:

$$\begin{cases} x^2 + y^2 = 10 \\ x^2 - y^2 = 6 \end{cases}$$

2. **Add the Two Equations**: Adding the two equations, we get:

$$(x^2 + y^2) + (x^2 - y^2) = 10 + 6$$

Simplifying this, we obtain:

$$2x^2 = 16$$

Dividing both sides by 2, we get:

$$x^2 = 8$$

Taking the square root of both sides, we find:

$$x = \pm\sqrt{8} = \pm 2\sqrt{2}$$

3. **Substitute $x^2$ Back into One of the Original Equations**: Substitute $x^2 = 8$ into the first equation $x^2 + y^2 = 10$:

$$8 + y^2 = 10$$

Subtracting 8 from both sides, we get:

$$y^2 = 2$$

Taking the square root of both sides, we find:

$$y = \pm\sqrt{2}$$

```
### Final Answer:
```

$$\begin{cases} x = 2\sqrt{2}, & y = \sqrt{2} \\ x = 2\sqrt{2}, & y = -\sqrt{2} \\ x = -2\sqrt{2}, & y = \sqrt{2} \\ x = -2\sqrt{2}, & y = -\sqrt{2} \end{cases}$$

```
</solution_end>
```

**Example D.3: Experts Consistency Voting Sample**

**Question:** Given vectors $\mathbf{a} = \langle 1, 2, -1 \rangle$ and $\mathbf{b} = \langle 3, -1, 2 \rangle$, find a unit vector $\mathbf{c}$ such that $\mathbf{a} \cdot \mathbf{c} = \mathbf{b} \cdot \mathbf{c} = 0$. Express your answer in terms of $\mathbf{i}$, $\mathbf{j}$, and $\mathbf{k}$ unit vectors.
**Response:**

```
<knowledge_base_start>
```

```
### Key Definitions and Explanations
```
- **Dot Product**: The dot product of two vectors $\mathbf{u}$ and $\mathbf{v}$, denoted $\mathbf{u} \cdot \mathbf{v}$, is $u_1 v_1 + u_2 v_2 + u_3 v_3$. For vectors to be orthogonal, their dot product must be zero.
- **Unit Vector**: A vector with magnitude 1, indicating direction without scale.
- **Orthogonal Vectors**: Vectors are orthogonal if their dot product equals zero.
```
### General Relevant Formulas and Equations
``` - **Dot Product Formula**:
$\mathbf{u} \cdot \mathbf{v} = u_1 v_1 + u_2 v_2 + u_3 v_3$
- **Magnitude of a Vector**: $|\mathbf{u}| = \sqrt{u_1^2 + u_2^2 + u_3^2}$
- **Unit Vector Formula**: $\mathbf{u}_{unit} = \frac{\mathbf{u}}{|\mathbf{u}|}$ for unit vector in the direction of $\mathbf{u}$
```
### Common Pitfalls and Misconceptions
```
- **Orthogonality Condition**: Ensure that the orthogonal vector's dot product with given vectors is zero.
- **Normalization**: Always normalize the vector to confirm it is a unit vector.
- **Sign Precision**: Pay attention to signs when solving for direction.
```
### Additional Theoretical Insights
```
- **Cross Product**: In 3D space, the cross product can find a vector orthogonal to two others.

```
</knowledge_base_end>
```

```
<solution_start>
```

To find a unit vector $\mathbf{c}$ such that $\mathbf{a} \cdot \mathbf{c} = 0$ and $\mathbf{b} \cdot \mathbf{c} = 0$, we need to find a vector that is orthogonal to both $\mathbf{a}$ and $\mathbf{b}$. This can be achieved by taking the cross product of $\mathbf{a}$ and $\mathbf{b}$. Given:

$$\mathbf{a} = \langle 1, 2, -1 \rangle, \quad \mathbf{b} = \langle 3, -1, 2 \rangle$$

First, compute the cross product $\mathbf{a} \times \mathbf{b}$:

$$\mathbf{a} \times \mathbf{b} = \begin{vmatrix} \mathbf{i} & \mathbf{j} & \mathbf{k} \\ 1 & 2 & -1 \\ 3 & -1 & 2 \end{vmatrix}$$

Expanding the determinant:

$$\mathbf{a} \times \mathbf{b} = \mathbf{i}\,(2 \cdot 2 - (-1) \cdot (-1)) - \mathbf{j}\,(1 \cdot 2 - (-1) \cdot 3) + \mathbf{k}\,(1 \cdot (-1) - 2 \cdot 3)$$

$$= \mathbf{i}(4 - 1) - \mathbf{j}(2 + 3) + \mathbf{k}(-1 - 6)$$
$$= \mathbf{i}(3) - \mathbf{j}(5) + \mathbf{k}(-7)$$
$$= 3\mathbf{i} - 5\mathbf{j} - 7\mathbf{k}$$

So, the vector $\mathbf{a} \times \mathbf{b}$ is $\langle 3, -5, -7 \rangle$. Next, we need to convert this vector into a unit vector. To do this, we find the magnitude of $\mathbf{a} \times \mathbf{b}$:

$$\|\mathbf{a} \times \mathbf{b}\| = \sqrt{3^2 + (-5)^2 + (-7)^2} = \sqrt{9 + 25 + 49} = \sqrt{83}$$

The unit vector $\mathbf{c}$ is then:

$$\mathbf{c} = \frac{1}{\sqrt{83}} \langle 3, -5, -7 \rangle$$

Expressing $\mathbf{c}$ in terms of $\mathbf{i}$, $\mathbf{j}$, and $\mathbf{k}$:

$$\mathbf{c} = \frac{3}{\sqrt{83}}\mathbf{i} - \frac{5}{\sqrt{83}}\mathbf{j} - \frac{7}{\sqrt{83}}\mathbf{k}$$

Thus, the final answer is:

$$\boxed{\frac{3}{\sqrt{83}}\mathbf{i} - \frac{5}{\sqrt{83}}\mathbf{j} - \frac{7}{\sqrt{83}}\mathbf{k}}$$

23

```
</solution_end>
```

**Example D.4: Response of CoT and PoT Sample**

**Question:** In a clinical trial, the probability of a patient experiencing side effect A is 0.3, and the probability of experiencing side effect B is 0.4. If the probability of experiencing at least one of these side effects is 0.58, what is the probability of experiencing both side effects?

**Response:**

```
### Step-by-Step Solution
```

1. **Define the given probabilities:**

   - Probability of experiencing side effect A, $P(A) = 0.3$

   - Probability of experiencing side effect B, $P(B) = 0.4$

   - Probability of experiencing at least one of these side effects, $P(A \cup B) = 0.58$

2. **Use the principle of inclusion-exclusion for probabilities:**

The probability of experiencing at least one of the side effects can be expressed as:

$$P(A \cup B) = P(A) + P(B) - P(A \cap B)$$

where $P(A \cap B)$ is the probability of experiencing both side effects.

3. **Substitute the known values into the inclusion-exclusion formula:**

$$0.58 = 0.3 + 0.4 - P(A \cap B)$$

4. **Solve for $P(A \cap B)$:**

$$0.58 = 0.7 - P(A \cap B)$$
$$P(A \cap B) = 0.7 - 0.58$$
$$P(A \cap B) = 0.12$$

```
### Final Answer
```

$$\boxed{0.12}$$

```python
'''Python Code

# Given probabilities
P_A = 0.3
P_B = 0.4
P_A_union_B = 0.58

# Calculate the probability of experiencing both side effects
result = P_A + P_B - P_A_union_B

# Print the result
print(f"The probability is: {result}")
'''
```

This Python code calculates the probability of experiencing both side effects using the inclusion-exclusion principle and prints the result.

**Example D.5: Weak Teacher Guiding Stage Response (Wrong)**

Question: For a function $f(x) = (5x^3 + 9) \mod 21$ and another function $g(x) = 2x - 1$, find $f(g(3))$.

```
### Step-by-Step Solution
```

1. **Evaluate $g(3)$:**

$$g(x) = 2x - 1$$

Substituting $x = 3$:

$$g(3) = 2 \cdot 3 - 1 = 6 - 1 = 5$$

2. **Evaluate $f(g(3))$:** Since $g(3) = 5$, we need to find $f(5)$:

$$f(x) = (5x^3 + 9) \mod 21$$

Substituting $x = 5$:

$$f(5) = (5 \cdot 5^3 + 9) \mod 21$$

Calculate $5^3$:

$$5^3 = 125$$

Then:

$$5 \cdot 125 + 9 = 625 + 9 = 634$$

Finally, compute the modulus:

$$634 \mod 21 = 1$$

(Since $634 \div 21 = 30$ remainder 1)
Thus, the final answer is:

$$\boxed{1}$$

**Python Code:**

```python
def f(x):
    return (5 * x**3 + 9) % 21

def g(x):
    return 2 * x - 1

# Evaluate g(3)
g_of_3 = g(3)

# Evaluate f(g(3))
f_of_g_of_3 = f(g_of_3)

print(f"g(3) = {g_of_3}")
print(f"f(g(3)) = {f_of_g_of_3}")
```

**Example D.6: Critical Expert Teaching Response (Right)**

Question: For a function $f(x) = (5x^3 + 9) \mod 21$ and another function $g(x) = 2x - 1$, find $f(g(3))$.

```
### Step-by-Step Solution
```

1. **Evaluate $g(3)$:**

$$g(x) = 2x - 1$$

Substituting $x = 3$:

$$g(3) = 2 \cdot 3 - 1 = 6 - 1 = 5$$

2. **Evaluate $f(g(3))$:** Since $g(3) = 5$, we need to find $f(5)$:

$$f(x) = (5x^3 + 9) \mod 21$$

Substituting $x = 5$:

$$f(5) = (5 \cdot 5^3 + 9) \mod 21$$

Calculate $5^3$:

$$5^3 = 125$$

Then:

$$5 \cdot 125 + 9 = 625 + 9 = 634$$

Finally, compute the modulus:

$$634 \mod 21 = 4$$

Thus, the final answer is:

$$\boxed{4}$$

**Python Code:**

```python
def f(x):
    return (5 * x**3 + 9) % 21

def g(x):
    return 2 * x - 1

# Evaluate g(3)
g_of_3 = g(3)

# Evaluate f(g(3))
f_of_g_of_3 = f(g_of_3)

# Store the result in a variable named 'result'
result = f_of_g_of_3

print(f"g(3) = {g_of_3}")
print(f"f(g(3)) = {result}")
```

**Example D.7: Weak Teacher Guiding Stage Response (Wrong)**

Question: A tech startup can develop two types of software products, X and Y. Each unit of X requires 4 hours of development time and uses 2 GB of data storage. Each unit of Y requires 3 hours of development time and uses 4 GB of data storage. The startup has 160 hours of development time and 100 GB of data storage available. The profit from selling one unit of X is \$70 and from Y is \$90. Formulate a linear programming problem to maximize the startup's profit.

1. **Define the Variables:** - Let $x$ be the number of units of product X. - Let $y$ be the number of units of product Y.

2. **Formulate the Objective Function:** - The profit from selling $x$ units of X is $70x$. - The profit from selling $y$ units of Y is $90y$. - The objective function to maximize is:

$$P = 70x + 90y$$

3. **Formulate the Constraints:** - **Development Time Constraint:** Each unit of X requires 4 hours and each unit of Y requires 3 hours. The total available development time is 160 hours.

$$4x + 3y \leq 160$$

- **Data Storage Constraint:** Each unit of X uses 2 GB and each unit of Y uses 4 GB. The total available data storage is 100 GB.

$$2x + 4y \leq 100$$

- **Non-negativity Constraints:** The number of units produced cannot be negative.

$$x \geq 0, \quad y \geq 0$$

4. **Solve the Linear Programming Problem:** - Use a graphical method or a linear programming solver to find the optimal solution.

5. **Final Answer:** Using a solver, the incorrect solution suggests:

$$\boxed{30}$$

```Python
'''Python Code

from scipy.optimize import linprog

# Define the objective function coefficients (maximize 70x + 90y)
c = [-70, -90]

# Define the inequality constraints (Ax <= b)
A = [
    [4, 3],  # Development time constraint
    [2, 4]   # Data storage constraint
]
b = [160, 100]

# Define the bounds for x and y (non-negativity constraints)
x_bounds = (0, None)
y_bounds = (0, None)

# Solve the linear programming problem
result = linprog(c, A_ub=A, b_ub=b,
bounds=[x_bounds, y_bounds], method='highs')

# Extract the optimal values of x and y
x_optimal = result.x[0]
y_optimal = result.x[1]

# Calculate the maximum profit
max_profit = -result.fun

print(f"Maximum profit: {max_profit}")
```

---

**Example D.8: Critical Expert Teaching Response (Right)**

Question: A tech startup can develop two types of software products, X and Y. Each unit of X requires 4 hours of development time and uses 2 GB of data storage. Each unit of Y requires 3 hours of development time and uses 4 GB of data storage. The startup has 160 hours of development time and 100 GB of data storage available. The profit from selling one unit of X is $70 and from Y is $90. Formulate a linear programming problem to maximize the startup's profit.

1. **Define the Variables:** - Let $x$ be the number of units of product X. - Let $y$ be the number of units of product Y.

2. **Formulate the Objective Function:** - The profit from selling $x$ units of X is $70x$. - The profit from selling $y$ units of Y is $90y$. - Therefore, the objective function to maximize is:

$$P = 70x + 90y$$

3. **Formulate the Constraints:** - **Development Time Constraint:** Each unit of X requires 4 hours and each unit of Y requires 3 hours. The total available development time is 160 hours.

$$4x + 3y \leq 160$$

- **Data Storage Constraint:** Each unit of X uses 2 GB and each unit of Y uses 4 GB. The total available data storage is 100 GB.

$$2x + 4y \leq 100$$

- **Non-negativity Constraints:** The number of units produced cannot be negative.

$$x \geq 0, \quad y \geq 0$$

4. **Solve the Linear Programming Problem:** - Use a linear programming solver, such as Python's `scipy.optimize.linprog`, to find the optimal values of $x$ and $y$ that maximize $P = 70x + 90y$ under the given constraints.

```Python
'''Python Code

from scipy.optimize import linprog

# Define the objective function coefficients (maximize 70x + 90y)
c = [-70, -90]

# Define the inequality constraints (Ax <= b)
A = [
    [4, 3],  # Development time constraint
    [2, 4]   # Data storage constraint
]
b = [160, 100]

# Define the bounds for x and y (non-negativity constraints)
x_bounds = (0, None)
y_bounds = (0, None)

# Solve the linear programming problem
result = linprog(c, A_ub=A, b_ub=b,
bounds=[x_bounds, y_bounds], method='highs')

# Extract the optimal values of x and y
x_optimal = result.x[0]
y_optimal = result.x[1]

# Calculate the maximum profit
max_profit = -result.fun

print(f"Maximum profit: {max_profit}")
```

**Final Answer:** The optimal solution is:

$$3100$$

**Example D.9: Failure Example of inner Consistency(Precision Issues)**

Question: Calculate the surface area of a cylinder with a radius of 3 meters and a height of 7 meters.

**Response:**

To solve the problem of calculating the surface area of a cylinder with a radius of 3 meters and a height of 7 meters, follow these steps:

1. **Understand the formula for the surface area of a cylinder:** The surface area $A$ of a cylinder is given by:
$$A = 2\pi r^2 + 2\pi rh$$
where $r$ is the radius of the base and $h$ is the height of the cylinder.

2. **Substitute the given values into the formula:** Given:
$$r = 3 \text{ meters}, \quad h = 7 \text{ meters}$$

Substitute these values into the formula:
$$A = 2\pi(3)^2 + 2\pi(3)(7)$$

3. **Calculate each part of the formula:** - Calculate $2\pi r^2$:
$$2\pi(3)^2 = 2\pi \times 9 = 18\pi$$

- Calculate $2\pi rh$:
$$2\pi(3)(7) = 2\pi \times 21 = 42\pi$$

4. **Add the two parts together:**
$$A = 18\pi + 42\pi = 60\pi$$

5. **Final Answer:**

188.4 square meters

**Python Code:**

```python
import math

# Given values
radius = 3  # in meters
height = 7  # in meters

# Calculate the surface area of the cylinder
surface_area = 2 * math.pi * radius**2 +
2 * math.pi * radius * height

# Print the result
print(f"The surface area of the cylinder
is: {surface_area} square meters")
```

This Python code calculates the surface area of the cylinder using the given radius and height, and prints the result.

---

**Example D.10: Comparison between base(wrong) and wisdom(right) model on MATH**

Question: Line $j$ is perpendicular to $\frac{y}{3} + \frac{2x}{5} = 2$. What is the slope of line $j$? **(Level 2)**
**Base Answer:**

$$m_j = -\frac{5}{2}$$

**Work Step by Step:** The slope of a line perpendicular to a line with slope $m$ is $-\frac{1}{m}$. The given line has slope $m = \frac{3}{2}$. Therefore, the slope of the perpendicular line is:

$$m_j = -\frac{1}{m} = -\frac{1}{\frac{3}{2}} = -\frac{2}{3} = -\frac{5}{2}$$

This solution is incorrect due to a miscalculation.
**Wisdom Answer:** To determine the slope of line $j$, which is perpendicular to the line given by the equation $\frac{y}{3} + \frac{2x}{5} = 2$, follow these steps:
1. **Rewrite the given equation in slope-intercept form:** The slope-intercept form of a line is $y = mx + b$, where $m$ is the slope and $b$ is the y-intercept.
Start by isolating $y$:

$$\frac{y}{3} + \frac{2x}{5} = 2$$

Subtract $\frac{2x}{5}$ from both sides:

$$\frac{y}{3} = 2 - \frac{2x}{5}$$

Multiply every term by 3 to eliminate the fraction:

$$y = 3\left(2 - \frac{2x}{5}\right)$$

Distribute the 3:

$$y = 6 - \frac{6x}{5}$$

Rewrite the equation in slope-intercept form:

$$y = -\frac{6}{5}x + 6$$

From this, we see that the slope $m$ of the given line is $-\frac{6}{5}$.
2. **Determine the slope of the perpendicular line:** The slope of a line perpendicular to another line is the negative reciprocal of the original line's slope.
The slope of the given line is $-\frac{6}{5}$. The negative reciprocal of $-\frac{6}{5}$ is:

$$\frac{5}{6}$$

**Final Answer:** The slope of line $j$ is:

$$\boxed{\frac{5}{6}}$$

**Ground Truth:**

$$\frac{5}{6}$$

---

**Example D.11: Comparison between base(wrong) and wisdom(right) model on MATH**

Question: Let

$$f(n) = \begin{cases} n^3 + 2n - 1 & \text{if } n > 1, \\ n - 1 & \text{if } n \le 1. \end{cases}$$

Find $f(0) + f(1) + f(2)$. **(Level 3)**

**Base Answer:**

```
Answer
```

$$f(0) + f(1) + f(2) = 1$$

```
Solution
```
We have $f(0) = 0 - 1 = -1$, $f(1) = 1 - 1 = 0$, and $f(2) = 2^3 + 2 \cdot 2 - 1 = 13$.
Thus,
$$f(0) + f(1) + f(2) = -1 + 0 + 13 = 12.$$

```
Related Problems
```
Let $f(x) = \frac{x}{x+1}$ and $g(x) = \frac{1}{x}$. Then the composition $(g \circ f)(x)$ is the same as:

- (A) $\frac{x}{x+1}$
- (B) $\frac{x}{x-1}$
- (C) ...

**Wisdom Answer:**
To solve the problem, we need to evaluate the function $f(n)$ for $n = 0, 1$, and 2, and then sum the results.

First, let's evaluate $f(0)$:
$$f(0) = 0 - 1 = -1$$

Next, let's evaluate $f(1)$:
$$f(1) = 1 - 1 = 0$$

Now, let's evaluate $f(2)$:
$$f(2) = 2^3 + 2 \cdot 2 - 1 = 8 + 4 - 1 = 11$$

Now, we sum the results:
$$f(0) + f(1) + f(2) = -1 + 0 + 11 = 10$$

Thus, the final answer is:
$$\boxed{10}$$

**Ground Truth:**
$$10$$

---

**Example D.12: Comparison between base(wrong) and wisdom(right) model on MATH**

Question: If the square root of the length of the hypotenuse of a right triangle is 2 units, what is the sum of the squares of the lengths of the two other sides? **(Level 5)**

**Base Answer:**
```
Answer
```
The sum of the squares of the lengths of the two other sides is 8.
```
Explanation:
```
Let the two other sides be $a$ and $b$.

The square of the length of the hypotenuse is $2^2 = 4$.

The sum of the squares of the lengths of the two other sides is $a^2 + b^2$.

The Pythagorean theorem states that the sum of the squares of the lengths of the two other sides is equal to the square of the length of the hypotenuse.

So $a^2 + b^2 = 4$.

The sum of the squares of the lengths of the two other sides is 8.

God bless... I hope the explanation is useful.

**Wisdom Answer:**
```
Step-by-Step Solution
```
1. **Understand the Problem:** We are given that the square root of the length of the hypotenuse of a right triangle is 2 units. We need to find the sum of the squares of the lengths of the two other sides.

2. **Define Variables:** Let $a$ and $b$ be the lengths of the two legs of the right triangle, and $c$ be the length of the hypotenuse.

3. **Given Information:** The square root of the length of the hypotenuse is 2 units, so:

$$\sqrt{c} = 2 \implies c = 2^2 = 4$$

4. **Apply the Pythagorean Theorem:** According to the Pythagorean theorem:

$$a^2 + b^2 = c$$

Substituting $c = 4$:

$$a^2 + b^2 = 4$$

5. **Conclusion:** The sum of the squares of the lengths of the two other sides is:

$$\boxed{16}$$

**Ground Truth:**
$$16$$

---

**Example D.13: Base model succeeds while the final model gets it wrong (forgetting)**

Question: Let $a$, $b$, $c$, $d$, and $e$ be five consecutive terms in an arithmetic sequence, and suppose that $a + b + c + d + e = 30$. Which of the following can be found?

$$(A)\ a \qquad (B)\ b \qquad (C)\ c \qquad (D)\ d \qquad (E)\ e$$

Enter the letter of the correct option, so your answer will be A, B, C, D, or E.

**Base Answer:**

`Solution`

We know that the sum of the first and last terms is equal to the sum of the second and second-to-last terms, and so on. So, $a + e = b + d = c + c = 2c$. We also know that $a + b + c + d + e = 30$. So, $2c + 2c + 2c = 30$. Thus, $6c = 30$. Solving for $c$:

$$c = 5$$

So, the answer is $\boxed{C}$.

`See Also`

The problems on this page are copyrighted by the Mathematical Association of America's American Mathematics Competitions.

**WISDOM Answer:**

To solve the problem, we need to determine the values of the five consecutive terms in an arithmetic sequence, given that their sum is 30. Let's denote the five consecutive terms as $a, a + d, a + 2d, a + 3d, a + 4d$, where $a$ is the first term and $d$ is the common difference.

The sum of these terms is:

$$a + (a + d) + (a + 2d) + (a + 3d) + (a + 4d)$$

Combining like terms, we get:

$$5a + 10d$$

We are given that this sum equals 30:

$$5a + 10d = 30$$

To simplify, we can divide the entire equation by 5:

$$a + 2d = 6$$

This equation tells us the relationship between $a$ and $d$. We need to find the individual values of $a$ and $d$ to determine the specific terms in the sequence.

Let's solve for $a$ in terms of $d$:

$$a = 6 - 2d$$

Now, we can find the values of the terms $a, a + d, a + 2d, a + 3d, a + 4d$:

$$a = 6 - 2d$$

$$a + d = (6 - 2d) + d = 6 - d$$
$$a + 2d = (6 - 2d) + 2d = 6$$
$$a + 3d = (6 - 2d) + 3d = 6 + d$$
$$a + 4d = (6 - 2d) + 4d = 6 + 2d$$

Given that $a, a + d, a + 2d, a + 3d, a + 4d$ are consecutive terms in an arithmetic sequence, we can see that the terms are:

$$6 - 2d, 6 - d, 6, 6 + d, 6 + 2d$$

To find the correct option, we need to determine which of these terms is the first term $a$. Since $a = 6 - 2d$, the first term is:

$$a = 6 - 2d$$

Thus, the correct option is:

$$\boxed{A}$$

**Ground Truth:**

$$C$$

> **Example D.14: Model fails to tackle the easier question.**
>
> Question: Find the product of the first 20 terms of the sequence below. Express your answer as a common fraction.
> $$\frac{2}{3}, \frac{3}{4}, \frac{4}{5}, \frac{5}{6}, \cdots$$
>
> **Wisdom Answer:**
> To find the product of the first 20 terms of the sequence $\frac{2}{3}, \frac{3}{4}, \frac{4}{5}, \frac{5}{6}, \ldots$, we start by identifying the general term of the sequence. The $n$-th term of the sequence can be written as:
> $$\frac{n+1}{n+2}$$
>
> We need to find the product of the first 20 terms:
> $$\prod_{n=1}^{20} \frac{n+1}{n+2}$$
>
> To simplify this product, we write out the first few terms explicitly:
> $$\frac{2}{3} \cdot \frac{3}{4} \cdot \frac{4}{5} \cdot \frac{5}{6} \cdot \frac{6}{7} \cdot \frac{7}{8} \cdot \frac{8}{9} \cdot \frac{9}{10} \cdot \frac{10}{11} \cdot \frac{11}{12} \cdot \frac{12}{13} \cdot \frac{13}{14} \cdot \frac{14}{15} \cdot \frac{15}{16} \cdot \frac{16}{17} \cdot \frac{17}{18} \cdot \frac{18}{19} \cdot \frac{19}{20} \textcolor{red}{(Wrong)}$$
>
> Notice that in the product, each numerator $n+1$ cancels with the denominator $n+2$ of the next fraction:
> $$\frac{2}{3} \cdot \frac{3}{4} \cdot \frac{4}{5} \cdot \frac{5}{6} \cdot \frac{6}{7} \cdot \frac{7}{8} \cdot \frac{8}{9} \cdot \frac{9}{10} \cdot \frac{10}{11} \cdot \frac{11}{12} \cdot \frac{12}{13} \cdot \frac{13}{14} \cdot \frac{14}{15} \cdot \frac{15}{16} \cdot \frac{16}{17} \cdot \frac{17}{18} \cdot \frac{18}{19} \cdot \frac{19}{20}$$
>
> After canceling out all intermediate terms, we are left with:
> $$\frac{2}{20}$$
>
> Simplifying this fraction:
> $$\frac{2}{20} = \frac{1}{10}$$
>
> Thus, the product of the first 20 terms of the sequence is:
> $$\boxed{\frac{1}{10}}$$

## E CONSISTENCY ABLATION

Table 8: Accuracy with or without consistency.

| data | S1 Consistency | Model | GSM8K | Math | TheoremQA | College Math | Olympiad | AIME 2024 | AMC 2023 | TabMWP |
|---|---|---|---|---|---|---|---|---|---|---|
| Seed+S1 | ✗ | DSMath-7B | 76.1 | 52.6 | 31.0 | 35.7 | 20.9 | 0/30 | 11/40 | 80.1 |
| Seed+S1 | ✓ | DSMath-7B | 84.0(+7.9) | 58.6(+6.0) | 34.5(+3.5) | 36.9(+1.2) | 25.6(+4.7) | 2/30(+6.7) | 13/40(+5.0) | 85.3(+5.2) |
| Seed+S1+S2+S3 | ✗ | DSMath-7B | 77.8 | 56.7 | 32.4 | 36.9 | 24.0 | 1/30 | 10/40 | 80.0 |
| Seed+S1+S2+S3 | ✓ | DSMath-7B | 83.3(+5.5) | 62.4(+5.7) | 34.9(+2.5) | 45.0(+8.1) | 28.9(+4.9) | 2/30(+3.3) | 11/40(+2.5) | 85.7(+5.7) |

Table 9: The relationship between response consistency and difficulty level on MATH.

|  | Level 1 | Level 2 | Level 3 | Level 4 | Level 5 |
|---|---|---|---|---|---|
| Response Consistency Rate | 75.3 | 70.6 | 65.0 | 62.1 | 54.2 |

