# OpenReview forum: "WISDOM: Progressive Curriculum Synthesis Makes LLMs Better Mathematical Reasoner"
_ICLR.cc/2025/Conference — Submitted to ICLR 2025_

### Official Review · Reviewer_azz3 · 2024-10-18

**Soundness:** 3
**Presentation:** 3
**Contribution:** 3
**Rating:** 6
**Confidence:** 4

**Summary:**

This work applies curriculum learning to guide LLMs in generating synthetic data through the Easy to Hard Cyclic Iterative Process. Based on that, this paper presents a three-stage framework "WISDOM".

**Strengths:**

1. Unlikely many contemporary works that scale up carelessly, this framework does attempt to be cost effective, mainly by using a weak and cheap teacher model first to solve many simple problems.

2. According to the evaluation results reported, it seems that the synthesized data are of good quality and does help models learn both in-domain and out-of-domain. This generality further proves the effectiveness of their method.

3. The results and analysis are very thorough. The authors not only provided main results but also discussed several factors and how they impact the learning performance.

**Weaknesses:**

1. In the methodology (Sec. 2.2), it seems that internal consistency (which is obtained as whether the model outputs the same answer via CoT and PoT) has been used as the only indicator of whether the model's response should be treated as correct. No ground truth labels are used here. I think it is quite evident that the self consistency of a single model under two prompting methods is far from a fair replacement of a ground truth. I think either this part of methodology is a critical weakness, or much more analysis is needed here. To be extra clear, I do buy the claim that there is probably a correlation between internal consistency and final accuracy, yet the gap between the two still looks too big for me without further justifications.

2. In the experiments, a number of baseline models are compared, on a set of data. However, the models have different data cutoff dates. Data contamination may be a possibility for some of the models, making the evaluation less fair. I feel data contamination is a very important aspect that worth being analyzed here.

3. A suggestion for the presentation: WISDOM is a good name, but I feel it has been used in a mixed way throughout the paper, which seems confusing at times. In Sec. 2, WISDOM has been used mainly as the framework that synthesizes data, while later in the experiment section, WISDOM is also used to denote the model. I recommend the authors to be clear and consistent with terms in the presentation, especially when it comes to core contributions like this.

**Questions:**

1. In the internal consistency part, why CoT and PoT are selected as the only two methods to assert consistency?

2. I feel the general iteration strategy looks similar to some more standard methods such as expert iteration in RL. It would be great to have some discussions about the connections, and what specifically in the framework that this paper presents is completely new.

3. As more and more data are generated, is there any negative tendencies worth noting? E.g. problem quality drop or diversity drop? More generally, is there any anticipated challenges when scaling the framework up?

**Details Of Ethics Concerns:**

N/A.

---

> ### Author Response · Authors · 2024-11-17
>
> # Response to Reviewer azz3
>
> We appreciate the reviewer’s detailed comments and thoughtful feedback. We have identified some misunderstandings that may have led to certain concerns and have addressed them in the responses below to provide clarification.
>
> **Summary of our method**:
>
> We adopt a progressive approach to enhance the difficulty of **newly synthesized problems** by leveraging the internal consistency of weak models, the consistency between strong and weak models, and the consistency of strong models.
> Furthermore, we expand diversity by incorporating a knowledge base enriched with meta-data.
> Unlike existing methods, such as Rejection Sampling and Expert Iteration, which primarily enhance the quantity and diversity of data to a limited extent, these methods rely on the existing ground truth within seed data for synthesis and do not generate entirely new questions.
> **This reliance constrains both the diversity and difficulty of the generated problems, makes them impossible for newly synthesized problems lacking ground truth, and restricts their applicability to broader scenarios.**
>
> Our research demonstrates（See Table 6, Appendix Table 8 and Appendix Table 9 in revision version）that employing **response consistency** for data synthesis and instruction difficulty evolution not only enables the unsupervised generation of a substantial number of new problems but also effectively enhances problem difficulty, response accuracy, and cost efficiency in these new problems.
>
>
> > W1: In the methodology (Sec. 2.2), it seems that internal consistency (which is obtained as whether the model outputs the same answer via CoT and PoT) has been used as the only indicator of whether the model's response should be treated as correct. No ground truth labels are used here. I think it is quite evident that the self consistency of a single model under two prompting methods is far from a fair replacement of a ground truth. I think either this part of methodology is a critical weakness, or much more analysis is needed here. To be extra clear, I do buy the claim that there is probably a correlation between internal consistency and final accuracy, yet the gap between the two still looks too big for me without further justifications.
>
> **Response**: The reviewer may have some misunderstandings regarding our data synthesis approach. First, **our method is not intended to replace ground truth but is necessary because the synthesized new questions inherently lack ground truth or true labels.**
> Second, as shown in the Appendix B.4, we generate CoT+PoT results **using a single prompt**, rather than employing multiple prompting methods as commonly seen in techniques like Self-Consistency. Thus, our approach is referred to as inner response consistency, emphasizing consistency based on a single prompt rather than cross-prompt consistency.
>
> Regarding the explanation of internal consistency and final accuracy, **due to the absence of ground truth**, we cannot directly compute final accuracy in its conventional sense.
> Instead, our method calculates the consistency between the inner response consistency of weak models and the output results of strong models to validate the effectiveness of inner response consistency in improving final accuracy.
>
> As for the relatively low consistency rates in Table 6, the primary reason lies in the high challenge level of the synthesized problems. For instance, on challenging benchmarks such as College Math and Olympiad Bench, even strong models like Qwen2-72B-instruct struggle to achieve an accuracy of 50%. This indicates that the output of strong models is not always correct. Furthermore, even if the results of strong and weak models are consistent, it does not necessarily mean that both are correct; it only reflects a trend of consistency between them. Therefore, our consistency-based evaluation focuses more on observing the potential impact of the data synthesis approach on difficulty and quality rather than solely relying on accuracy as the evaluation metric.
>
> > W2: In the experiments, a number of baseline models are compared, on a set of data. However, the models have different data cutoff dates. Data contamination may be a possibility for some of the models, making the evaluation less fair. I feel data contamination is a very important aspect that worth being analyzed here.
>
> **Response**: Thank you for your suggestion regarding data contamination; we fully agree on the importance of this aspect. While we are unsure whether data contamination checks were performed for other baseline models, **we have thoroughly analyzed the potential contamination in our dataset, as detailed in Appendix A.2.** Furthermore, we have implemented appropriate measures to ensure the fairness of our experiments and the reliability of our data. Once again, we appreciate your suggestion, as it has been very helpful in improving the rigor of our work.

---

> > ### Author Response · Authors · 2024-11-17
> >
> > > W3: A suggestion for the presentation: WISDOM is a good name, but I feel it has been used in a mixed way throughout the paper, which seems confusing at times. In Sec. 2, WISDOM has been used mainly as the framework that synthesizes data, while later in the experiment section, WISDOM is also used to denote the model. I recommend the authors to be clear and consistent with terms in the presentation, especially when it comes to core contributions like this.
> >
> > **Response**: The WISDOM datasets are synthesized datasets, while the WISDOM models are fine-tuned on these datasets.
> > Additionally, WISDOM itself serves as a methodological framework.
> > This naming convention aligns with the terminology used in other baseline methods [1-4].
> >
> > > Q1: In the internal consistency part, why CoT and PoT are selected as the only two methods to assert consistency?
> >
> > **Response**:In the field of mathematics, Chain of Thought (CoT) and Program of Thought (PoT) are among the most classical and state-of-the-art methods for improving model performance[5-6], which is why we chose to adopt them in our work.
> > However, our proposed approach is not limited to CoT and PoT. Due to its versatility, our method can be easily adapted to other similar approaches.
> >
> > > Q2: I feel the general iteration strategy looks similar to some more standard methods such as expert iteration in RL. It would be great to have some discussions about the connections, and what specifically in the framework that this paper presents is completely new.
> >
> > **Response**: Our approach differs from expert iteration and the STaR method mentioned by Reviewer q9o3 primarily in its focus on dynamic instruction difficulty evolution, whereas expert iteration emphasizes dynamic model iteration evolution. Unlike expert iteration, our method does not involve policy updates and does not fall under the domain of reinforcement learning. Instead, it is more focused on data generation. In contrast, expert iteration centers on the process of policy updates.
> >
> > Additionally, we efficiently implement instruction difficulty evolution through response consistency. Even in the absence of ground truth, our method significantly increases the difficulty and diversity of instructions, enhancing the capabilities of large language models (LLMs) in complex mathematical reasoning tasks.
> >
> > While expert iteration effectively increases the quantity and diversity of data, its limitation lies in its inability to generate entirely new problems.
> > All generated question-response pairs depend on the ground truth present in the seed data, and the generation process is largely an exploration of different response sampling paths.
> > This reliance not only restricts the diversity and difficulty of the problems but also narrows the applicability of the method in broader scenarios.
> >
> > In contrast, our approach can generate entirely novel problems without relying on ground truth. By dynamically adjusting through response consistency, we can progressively increase the difficulty of generated problems and improve response accuracy. Furthermore, by incorporating knowledge bases and meta-information, we enhance the diversity of synthesized problems, providing more comprehensive and flexible data support for complex reasoning tasks.
> >
> > > Q3: As more and more data are generated, is there any negative tendencies worth noting? E.g. problem quality drop or diversity drop? More generally, is there any anticipated challenges when scaling the framework up?
> >
> > **Response**: From Figures 7 and 8, it can be observed that as the data volume increases, the performance improvement of the model gradually exhibits a trend of diminishing marginal returns. This indicates that while synthesized data significantly enhances model performance in the initial stages, its incremental impact becomes less pronounced once the data reaches a certain scale. This phenomenon is common in data-driven learning tasks, reflecting the non-linear relationship between data scale and model performance.[1-4]
> >
> > [1] DART-Math: Difficulty-Aware Rejection Tuning for Mathematical Problem-Solving. NeurIPS 2024.
> > [2] METAMATH: BOOTSTRAP YOUR OWN MATHEMATICAL QUESTIONS FOR LARGE LANGUAGE MODELS. ICLR 2024.
> > [3] MAMMOTH: BUILDING MATH GENERALIST MODELS THROUGH HYBRID INSTRUCTION TUNING. ICLR 2024.
> > [4] MathScale: Scaling Instruction Tuning for Mathematical Reasoning. ICML 2024.
> > [5] TORA: A TOOL-INTEGRATED REASONING AGENT FOR MATHEMATICAL PROBLEM SOLVING. ICLR 2024.
> > [6] MATHCODER: SEAMLESS CODE INTEGRATION IN LLMS FOR ENHANCED MATHEMATICAL REASONING. ICLR 2024.

---

> > > ### Comment · Reviewer_azz3 · 2024-11-22
> > > **Reviewer response to author rebuttal**
> > >
> > > Thank the authors for the detailed responses. They do help clarify many aspects of this paper. Now regarding some specific points in the authors' response:
> > >
> > > ## Re W1:
> > >
> > > The use of inner-response consistency, unlike most other self-consistency papers, does make sense to me. I do understand that the ground truth is not available in this work, and this connects to why the authors are using consistency-based methods. This makes sense to me, yet if I understand correctly, the consistency does not, and cannot, be a substitute of ground truth -- it can instead arguably contribute to a better accuracy. Is this understanding correct? If so that makes sense now, and I think it would help to directly clarify this more in the paper too.
> > >
> > > ## Re W2:
> > >
> > > Glad to see the authors' efforts in analyzing the data contamination problem, thanks!
> > >
> > > ## Re the responses to my previous questions:
> > >
> > > Those explanations are great! Thank the authors for the additional efforts. For Q3 specifically, I think it would be helpful not only to note down the general trends which I think very much aligns with the authors' initial hypotheses, but also provide some representative qualitative examples directly -- e.g. when the initial model succeeds in a problem while the final model gets it wrong (forgetting); when the initial model fails on a problem yet the fine-tuned model gets it right (improvement); when the model successfully answers a question while getting an arguably easier question wrong.

---

> > > > ### Author Response · Authors · 2024-11-23
> > > >
> > > > Thank you for your thoughtful follow-up discussion and all the constructive suggestions provided. We deeply appreciate the time and effort you have dedicated to engaging with our work, as well as your insightful comments that have helped us refine both the clarity and impact of our paper. Please find our detailed responses to your latest comments below.
> > > >
> > > > > The use of inner-response consistency, unlike most other self-consistency papers, does make sense to me. I do understand that the ground truth is not available in this work, and this connects to why the authors are using consistency-based methods. This makes sense to me, yet if I understand correctly, the consistency does not, and cannot, be a substitute of ground truth -- it can instead arguably contribute to a better accuracy. Is this understanding correct? If so that makes sense now, and I think it would help to directly clarify this more in the paper too.
> > > >
> > > >
> > > > We sincerely thank the reviewer for recognizing the contributions of our use of inner response consistency.
> > > >
> > > > Currently, improving the mathematical reasoning capabilities of large language models requires high-quality, diverse, and challenging data. As highlighted in the introduction of our paper, the availability of high-quality data diminishes as training scales increase. Synthetic data can address this issue by generating entirely novel problems, thereby enhancing the generalization of model reasoning. However, without manual validation of each generated problem, it is challenging to ensure the correctness of the answers, and manual verification at the scale of millions or tens of millions of data points is both impractical and cost-prohibitive.
> > > >
> > > > Our method leverages response consistency **not only to progressively enhance the diversity and difficulty of newly generated problems but also to improve the accuracy of answers in the absence of ground truth**. The models fine-tuned on this synthesized dataset outperform lots of baselines and achieve sota performance.
> > > >
> > > > Your understanding is indeed correct: if newly synthesized problems come with ground truth answers, the accuracy of those answers would undoubtedly be the highest, and ground truth cannot be replaced. However, in the absence of ground truth, our approach of utilizing inner response consistency to improve the accuracy of responses and model performance is one of our key contributions.
> > > >
> > > > Additionally, we have revised **Figure 1 and text (line 67-78)** to more clearly illustrate this in the paper.

---

> > > > ### Author Response · Authors · 2024-11-23
> > > >
> > > > > For Q3 specifically, I think it would be helpful not only to note down the general trends which I think very much aligns with the authors' initial hypotheses, but also provide some representative qualitative examples directly -- e.g. when the initial model succeeds in a problem while the final model gets it wrong (forgetting); when the initial model fails on a problem yet the fine-tuned model gets it right (improvement); when the model successfully answers a question while getting an arguably easier question wrong.
> > > >
> > > >
> > > > We sincerely thank the reviewers for their insightful feedback, particularly regarding qualitative examples. In response, we have conducted a detailed comparison between the original DeepSeek base model and the fine-tuned DeepSeek WISDOM model on the WISDOM datasets. The results have been added to the appendix for your reference.
> > > >
> > > > +	**Forgetting**: As shown in Appendix D.13, the examples illustrate that the base model sometimes provides correct answers, likely due to its exposure to similar problems during pretraining, even including references to the original sources in its responses. However, after fine-tuning, the model’s mathematical reasoning capabilities are significantly enhanced, albeit at the cost of “forgetting” memorized problems, which constitute a negligible portion (less than 0.5%).
> > > >
> > > > +	**Improvement**: Please refer to Appendices D.10–D.12 for evidence of the improved reasoning and problem-solving capabilities of the fine-tuned model.
> > > >
> > > > +	**Instability**: Errors in solving simpler problems typically arise from mistakes in intermediate steps, as detailed in Appendix D.14, which propagate to incorrect final answers.
> > > >
> > > > We sincerely appreciate your understanding of our overall framework. Our approach highlights the critical role of inner response consistency in increasing problem difficulty and enhancing problem diversity during data synthesis. Notably, **inner response consistency also improves the accuracy of generated responses, even in the absence of ground truth answers**. Furthermore, models trained on this dataset achieve state-of-the-art performance not only on in-domain tasks but also across many challenging out-of-domain datasets. Additionally, we have contributed **millions of high-quality data points and models to the open-source community** to facilitate further research in this area.
> > > >
> > > > We hope our response has adequately addressed your concerns. If so, we kindly request you to consider revising your evaluation score. Thank you once again for your invaluable feedback and thoughtful consideration of our work.

---

> > > > > ### Author Response · Authors · 2024-11-25
> > > > >
> > > > > Dear Reviewer azz3,
> > > > >
> > > > > Thank you once again for your valuable comments and suggestions, which have been incredibly helpful in improving our work. We have provided detailed responses to the concerns raised and included additional experimental results to further address the points you highlighted.
> > > > >
> > > > > With the discussion period concluding in two days, **we kindly ask if you could let us know whether our responses have resolved your concerns. If there are any remaining questions or comments, we are eager to engage in further discussion and will do our best to address them promptly.**
> > > > >
> > > > > We truly appreciate your time and effort, especially during this busy period.
> > > > >
> > > > > Best regards,
> > > > >
> > > > > The Authors

---

> > > > > > ### Comment · Reviewer_azz3 · 2024-11-26
> > > > > >
> > > > > > I would like to thank the authors for their thorough responses and additional efforts. My questions have been sufficiently answered, and I believe with the clarifications and revisions, this work's presentation is much improved and its contribution is sound. I have raised both scores to reflect the improvement.
> > > > > >
> > > > > > I also raised my overall review score to reflect this. Thank the authors again for the efforts.

---

> > > > > > > ### Author Response · Authors · 2024-11-27
> > > > > > >
> > > > > > > Thank you for your thoughtful feedback and for taking the time to engage with our work so thoroughly.
> > > > > > > We greatly appreciate your detailed questions and insights, which helped us refine the clarity and presentation of our paper.
> > > > > > > We are also grateful for your recognition of the contributions to our work.
> > > > > > > Thank you once again for your time and constructive discussions.
> > > > > > >
> > > > > > > Bests,
> > > > > > >
> > > > > > > Authors

---

### Official Review · Reviewer_q9o3 · 2024-11-04

**Soundness:** 3
**Presentation:** 2
**Contribution:** 2
**Rating:** 6
**Confidence:** 3

**Summary:**

This paper introduces a synthetic data generation method for mathematical reasoning problems, and shows substantial gains for Llama and Qwen models when fine-tuning on their generated data. The process has 3 stages: one amounting to attempting to solve problems using Chain-of-Thought + Program-of-Thought, the second in using a "weak teacher" model to analyze solutions where CoT and PoT disagreed (and attempt to solve them again), and a third stage where an even stronger model does the same. With GSM8k and MATH as seed datasets, the authors observe substantial gains on MATH (e.g. +27% for Qwen72B) and other datasets not used for training, such as AMC2023, TheoremQA, and the AIME 2024 challenge (often leading to 1-2 extra problems solved, out of 30).

**Strengths:**

* Synthetic data generation is a very relevant direction to improve current LLMs on hard reasoning tasks
* While the individual ideas are generally not new (related to distillation, expert iteration, self-consistency), this seems to be a particular combination that works well
* The authors provide comprehensive experiments on several base LLMs and datasets. The gains are clear in several cases, and bring the larger open source models (at the 70B scale) close to closed generalist models, like Claude 3 and GPT-4o-0513.
* The authors ablated both the stages of data generation, the use of self-consistency, and the diversity-improving heuristic

**Weaknesses:**

* The pipeline is quite intricate, and I don't think the authors compared (or cited, if someone has done these before) to the simplest possible data generation methods. Two of these would be the Self-Taught Reasoner [1] method (essentially expert iteration, though it also has a rationalization step) and CoT distillation [2]. I'm not 100% convinced that the whole pipeline is necessary, since overall this amounts to distilling from self-consistency and from GPT-4o.
* The authors don't use the answers in the seed datasets. While this can be seen as a strength (it makes less assumptions), one has to wonder if it wasn't possible to do better, and cheaper, by using the dataset-provided answers, rather than self-consistency. It doesn't seem exactly valuable to me to make an artificial assumption that doesn't hold in either of the seed datasets used in the experiments. The more general pipeline only becomes compelling if you actually apply it to a source of problems that have no answers associated with them (e.g. extracted from more diverse Web sources, problem sets, etc), in which case methods like STaR directly don't apply (though STaR with self-consistency has already been tried, too [3]).
* [Minor] The narrative around curriculum learning is a bit inaccurate (compared to the traditional use of the term in Machine Learning [4]). While I understand there is supposedly an "implicit curriculum" in the questions that different iterations get right, this is not really guaranteed to align with any human-interpretable curriculum. LLMs have been shown to often solve hard questions and still fail to answer easy ones, if difficulty is defined by human curriculum. Curriculum learning generally means that you pre-define a sequence for the training set, according to some external labeling, not to where the model itself succeeds or fails.

[1] Star: Bootstrapping reasoning with reasoning. NeurIPS 2022
[2] Specializing Smaller Language Models towards Multi-Step Reasoning. ICML 2023
[3] Large Language Models Can Self-Improve. EMNLP 2023
[4] Curriculum learning. ICML 2009

**Questions:**

* What was the actual cost of the experiments? In 3.8.2 (Cost Saving), the authors mention the *saving*, but not the actual cost. It's important to get an absolute estimate, e.g. for each round of MATH on Llama 70B. What would be the comparative cost to just do distillation from GPT-4o directly, maybe with rationalization (as STaR did)?
* Do the authors have examples of a same question that the model initially gets wrong, but gets right after fine-tuning? It would be good to see a few qualitative examples and try to get a sense of how the model's behavior changes over time.

---

> ### Author Response · Authors · 2024-11-17
>
> # Response to Reviewer q9o3
>
> We appreciate the reviewer’s detailed comments and suggestive feedback. Upon review, we notice both general and specific misunderstandings that may have contributed to some of the concerns raised. We have addressed these points and provided clarifications in the responses below.
>
> **Summary of our method**:
>
> We adopt a progressive approach to enhance the difficulty of **newly synthesized problems** by leveraging the internal consistency of weak models, the consistency between strong and weak models, and the consistency of strong models.
> Furthermore, we expand diversity by incorporating a knowledge base enriched with meta-data.
> Unlike existing methods, such as Rejection Sampling and Expert Iteration, which primarily enhance the quantity and diversity of data to a limited extent, these methods rely on the existing ground truth within seed data for synthesis and do not generate entirely new questions.
> **This reliance constrains both the diversity and difficulty of the generated problems, makes them impossible for newly synthesized problems lacking ground truth, and restricts their applicability to broader scenarios.**
>
> Our research demonstrates（See Table 6, Appendix Table 8 and Appendix Table 9 in revision version）that employing **response consistency** for data synthesis and instruction difficulty evolution not only enables the unsupervised generation of a substantial number of new problems but also effectively enhances problem difficulty, response accuracy, and cost efficiency in these new problems.
>
> > W1: The pipeline is quite intricate, and I don't think the authors compared (or cited, if someone has done these before) to the simplest possible data generation methods. Two of these would be the Self-Taught Reasoner [1] method (essentially expert iteration, though it also has a rationalization step) and CoT distillation [2]. I'm not 100% convinced that the whole pipeline is necessary, since overall this amounts to distilling from self-consistency and from GPT-4o.
>
> **Response**:
> A key distinction between our approach and STaR[1] lies in our adoption of dynamic instruction difficulty evolution, as opposed to [1]’s expert iteration, which focuses on dynamic model iteration updates. Our method does not involve policy updating, model updating or reinforcement learning. Instead, we emphasize the data generation process, whereas expert iteration places more emphasis on policy updating.
>
> Furthermore, **we efficiently achieve instruction difficulty evolution through response consistency, including weak model and strong model.** This approach enables us to enhance the difficulty and diversity of instructions **even in the absence of ground truth**, in contrast, STaR entirely relies on existing ground truth for CoT data synthesis and does not generate new questions.
>
> Our method not only addresses the limitations of relying solely on self-consistency to distill GPT-4-generated data but also offers a more flexible and efficient strategy for difficulty control and fully leverage the strength of weak teacher model and strong expert model.
>
>
> > W2: The authors don't use the answers in the seed datasets. While this can be seen as a strength (it makes less assumptions), one has to wonder if it wasn't possible to do better, and cheaper, by using the dataset-provided answers, rather than self-consistency. It doesn't seem exactly valuable to me to make an artificial assumption that doesn't hold in either of the seed datasets used in the experiments. The more general pipeline only becomes compelling if you actually apply it to a source of problems that have no answers associated with them (e.g. extracted from more diverse Web sources, problem sets, etc), in which case methods like STaR directly don't apply (though STaR with self-consistency has already been tried, too [3]).
>
> **Response**:
> There might be some misunderstandings regarding our approach. While our method leverages response consistency to achieve difficulty evolution, **it also generates entirely new questions without ground truth in each iteration round. This aligns with the broader practice of extracting questions from external sources or repositories, where ground truth is similarly unavailable.**
>
> For these newly generated questions, we also apply response consistency to enhance both their difficulty and the quality of the responses. Experimental results (detailed in Table 6 and Table 8 in Appendix E) demonstrate that, compared to the methods that do not utilize consistency, our approach significantly improves the response accuracy for these questions.

---

> > ### Author Response · Authors · 2024-11-17
> >
> > > W3: [Minor] The narrative around curriculum learning is a bit inaccurate (compared to the traditional use of the term in Machine Learning [4]). While I understand there is supposedly an "implicit curriculum" in the questions that different iterations get right, this is not really guaranteed to align with any human-interpretable curriculum. LLMs have been shown to often solve hard questions and still fail to answer easy ones, if difficulty is defined by human curriculum. Curriculum learning generally means that you pre-define a sequence for the training set, according to some external labeling, not to where the model itself succeeds or fails.
> >
> > **Response**: The statement that “the model cannot solve simple problems” is problematic.
> > While LLMs may occasionally make errors on simple problems, this does not imply that they are incapable of solving such problems. In this paper, we do not design the difficulty mannually, and measure the difficulty of the problem based on the response consistency of model. If a model consistently fails on these problems, then these problems would not be categorized as “simple.” For instance, in the GSM8K dataset, which consists of elementary-level problems, LLMs occasionally make mistakes, resulting in accuracy below 100%. However, this does not indicate that they are unable to handle simple problems.
> >
> > **The success or failure of response consistency within the model is more relevant to dynamically adjusting and evolving the difficulty of instructions.** Our experiments further demonstrate that the model’s response consistency is effective in driving instruction difficulty evolution and generating higher-quality data.
> >
> > To verify the hypothesis "simpler problems are more likely to yield consistent results across various solutions, including CoT and PoT", we design experiments to explore the relationship between inner response consistency and problem difficulty. Using a specific prompt (see Appendix B.4) and the DeepSeek-V2.5 model (note: the V2.0 API has been deprecated due to updates), we test the relationship between response consistency and difficulty on 5,000 test problems from the MATH dataset. Difficulty is measured using the “Level” tags provided by the dataset. The results (see the below table, revised in Appendix Table 9) show a clear trend: as problem difficulty increases, response consistency decreases. This trend strongly suggests that response consistency is a simple yet effective metric for evaluating problem difficulty. Furthermore, as shown in Tables 6 and 8, we observe that response consistency positively contributes to improving mathematical reasoning performance, both in terms of accuracy and quality.
> >
> > |                           | Level 1 | Level 2 | Level 3 | Level 4 | Level 5 |
> > |---------------------------|---------|---------|---------|---------|---------|
> > | Response Consistency Rate | 75.3    | 70.6    | 65.0    | 62.1    | 54.2    |
> >
> > > Q1: What was the actual cost of the experiments? In 3.8.2 (Cost Saving), the authors mention the saving, but not the actual cost. It's important to get an absolute estimate, e.g. for each round of MATH on Llama 70B. What would be the comparative cost to just do distillation from GPT-4o directly, maybe with rationalization (as STaR did)?
> >
> > **Response**: Section 3.8.2 presents the actual costs incurred. The actual cost of experiments are shown in Figure 6, WISDOM cost 12068 USD. If just do distillation from GPT-4o directyly, it will cost 34002 USD. All the costs reported correspond to API usage expenses. "Our approach is 2.82 times more cost-effective compared to majority voting, leading to a total savings of over 20,000 US dollars in overall expenditure."
> >
> > > Q2: Do the authors have examples of a same question that the model initially gets wrong, but gets right after fine-tuning? It would be good to see a few qualitative examples and try to get a sense of how the model's behavior changes over time.
> >
> >
> > **Response**: It is important to clarify that during our data synthesis process, the dataset undergoes progressively dynamic iteration, with its difficulty progressively increasing and its size expanding. However, **the model for generating the data remains unchanged.** Ultimately, the dataset produced through this progressive curriculum learning approach is used to fine-tune other base models, resulting in the WISDOM Series Models.
> >
> > Notably, during the Weak Teacher Guiding Stage, some newly generated questions may contain errors or inconsistencies. However, in the subsequent Critical Expert Teaching Stage, these issues are addressed through further refinement and optimization, ensuring the questions become both consistent and accurate. Specific examples of this process are provided in the Appendix D.5-D.8.

---

> > > ### Comment · Reviewer_q9o3 · 2024-11-22
> > >
> > > Thanks for the clarifications. These do clear out most of my questions. It now makes sense that you do have to rely on self-consistency to get reliable answers to your newly synthesized questions. Two suggestions to clarify this:
> > >
> > > * In Figure 1, is "Hard Instruction Evolving" where the new questions are synthesized? This label was not very suggestive of that at first. I now went through the prompt in Appendix B.2, so assuming that's what's happening in this new box, I see it now. Perhaps the label can be made more direct.
> > > * Algorithm 1, line 27: I suppose here you wanted to take the union?
> > >
> > > > Response: The statement that “the model cannot solve simple problems” is problematic. [...]
> > >
> > > I appreciate the new experiment, but was is in quotes here is not what I said. All I meant was that it is possible for a model to simultaneously (a) successfully answer a "harder" question, according to a human-made curriculum, and (b) fail to answer an "easy" one. Your experiments indeed confirm that: the model fail at many Level 1 questions, and simultaneously answer many Level 5 questions correctly. (Of course, on average, it performs better on Level 1 than on Level 5).
> > >
> > > In any case, I still recommend revising the use of "curriculum learning" because this exact term, as used in the machine learning literature, tends to imply that you pre-specify the ordering of problems (e.g., see the paper I cited, or the many others that came since then). In your case, you define the "curriculum" based on the model's performance itself. This is a perfectly reasonable thing to do, it's just not what "curriculum learning" usually means. I just commented on the terminology for clarity.
> > >
> > > > Response: It is important to clarify that during our data synthesis process, the dataset undergoes progressively dynamic iteration, with its difficulty progressively increasing and its size expanding. However, the model for generating the data remains unchanged. Ultimately, the dataset produced through this progressive curriculum learning approach is used to fine-tune other base models, resulting in the WISDOM Series Models.
> > >
> > > Here, I was referring to evaluation problems, not the synthesized problems. I'd just want to understand better the capabilities that the model is gaining undergoing WISDOM training. Since performance improves, you must have several examples of evaluation problems (e.g., in MATH) that (a) the base model gets wrong, and (b) the fine-tuned model gets right. My question was about those - I think looking at several examples of those would be highly instructive.

---

> > > > ### Author Response · Authors · 2024-11-23
> > > >
> > > > We sincerely thank the reviewer for thorough evaluation and for providing detailed and insightful feedback. Their constructive comments on both the overall direction and specific details of the paper have been invaluable in helping us improve the clarity, quality, and impact of our work.
> > > >
> > > > > In Figure 1, is "Hard Instruction Evolving" where the new questions are synthesized? This label was not very suggestive of that at first. I now went through the prompt in Appendix B.2, so assuming that's what's happening in this new box, I see it now. Perhaps the label can be made more direct.
> > > >
> > > > “Hard Instruction Evolving” indeed refers to the synthesis of new problems as described in Appendix B.2. Our intention here is to highlight the contributions of our method, with a particular **focus on difficulty and diversity when synthesizing new problems**. In the revised version, we have made this aspect more explicitly and directly emphasized.
> > > > > Algorithm 1, line 27: I suppose here you wanted to take the union?
> > > >
> > > > This approach involves taking the union, aiming to **leverage the knowledge base of the questions as well as meta-information such as skills and topics** to diversify the generated problems as much as possible.
> > > >
> > > > > In any case, I still recommend revising the use of "curriculum learning" because this exact term, as used in the machine learning literature, tends to imply that you pre-specify the ordering of problems (e.g., see the paper I cited, or the many others that came since then). In your case, you define the "curriculum" based on the model's performance itself. This is a perfectly reasonable thing to do, it's just not what "curriculum learning" usually means. I just commented on the terminology for clarity.
> > > >
> > > > I sincerely appreciate your suggestions regarding the use of “curriculum learning.” Unlike its traditional usage in machine learning literature, where curriculum learning refers to the pre-defined sequencing of tasks, **our approach defines the “curriculum” based on the model’s performance**. We have addressed this distinction in the **revised version(line 68-71)** to ensure a clearer differentiation from the conventional terminology in machine learning.
> > > >
> > > > > Here, I was referring to evaluation problems, not the synthesized problems. I'd just want to understand better the capabilities that the model is gaining undergoing WISDOM training. Since performance improves, you must have several examples of evaluation problems (e.g., in MATH) that (a) the base model gets wrong, and (b) the fine-tuned model gets right. My question was about those - I think looking at several examples of those would be highly instructive.
> > > >
> > > > We provide **specific comparative examples** of the DeepSeek base model and the DeepSeek Wisdom model fine-tuned on the WISDOM dataset in **Appendix D.10-12** for your better assessment.
> > > >
> > > > Before fine-tuning, the base model exhibits several shortcomings: it lacks a clear reasoning path, generates answers that fail to terminate properly, and often makes simple computational errors. In contrast, the model fine-tuned on the WISDOM dataset demonstrates clear reasoning paths and consistently produces correct results. This indicates that the model fine-tuned on our synthesized dataset achieves significantly improved mathematical reasoning capabilities.
> > > >
> > > > > It now makes sense that you do have to rely on self-consistency to get reliable answers to your newly synthesized questions.
> > > >
> > > > We sincerely appreciate your understanding of our overall framework. Our approach **not only leverages self-consistency but also demonstrates the pivotal role of inner response consistency in upgrading the difficulty and enhancing the diversity of problems during data synthesis**. Furthermore, the models trained on this dataset achieve Sota performance both on in-domain tasks and across lots of challenging out-of-domain datasets. Additionally, we have contributed millions of data points and models to the open-source community.
> > > >
> > > > We hope our response has effectively addressed your concerns. If so, we kindly ask you to consider revisiting and potentially adjusting the evaluation score. Thank you for your thoughtful consideration.

---

> > > > > ### Author Response · Authors · 2024-11-25
> > > > >
> > > > > Dear Reviewer q9o3,
> > > > >
> > > > > Thank you once again for your valuable comments and suggestions, which have been incredibly helpful in improving our work. We have provided detailed responses to the concerns raised and included additional experimental results to further address the points you highlighted.
> > > > >
> > > > > With the discussion period concluding in two days, **we kindly ask if you could let us know whether our responses have resolved your concerns. If there are any remaining questions or comments, we are eager to engage in further discussion and will do our best to address them promptly.**
> > > > >
> > > > > We truly appreciate your time and effort, especially during this busy period.
> > > > >
> > > > > Best regards,
> > > > >
> > > > > The Authors

---

> > > > > > ### Comment · Reviewer_q9o3 · 2024-11-27
> > > > > >
> > > > > > Thanks for the response. I appreciate the examples, and the revisions to the paper. I have revised my score.
> > > > > >
> > > > > > Overall, the results are interesting in several cases (like Llama 3 8B getting almost 60% on MATH). The cost of the results was quite significant, though, so I'm skeptical whether other researchers will try to replicate or build on the pipeline directly. But it is still valuable to see how far you can push smaller model via distillation-based methods, and this paper shows an interesting data point in that direction. Methodologically, perhaps the main novelty here is combining distillation with the synthesis of novel problems, getting reliability via various consistency methods.
> > > > > >
> > > > > > (here, I believe the "expert teacher" model is still roughly an upper bound, which the current results support - I wouldn't expect the WISDOM student model to get beyond what the teacher is capable of, so that's why I'm understanding it as fundamentally a distillation approach).
> > > > > >
> > > > > > > This approach involves taking the union, aiming to leverage the knowledge base of the questions as well as meta-information such as skills and topics to diversify the generated problems as much as possible.
> > > > > >
> > > > > > If so, then shouldn't line 27 have an union? I'm referring to the end of the line, $\mathcal{Q} \leftarrow ... $
> > > > > >
> > > > > > Thanks again to the authors.

---

> > > > > > > ### Author Response · Authors · 2024-11-27
> > > > > > >
> > > > > > > We sincerely appreciate your detailed discussion, valuable suggestions, and recognition of the value of our work.
> > > > > > > > The cost of the results was quite significant, though, so I'm skeptical whether other researchers will try to replicate or build on the pipeline directly.
> > > > > > >
> > > > > > > Thank you for raising concerns regarding the cost and replicability of our approach, which align with our motivation to enhance **efficiency**. The costs appear relatively high because we aimed to explore two key aspects:
> > > > > > > + Observing whether scaling our synthesized dataset to millions of samples could still improve model performance, particularly since adding more data does not always result in better performance and can sometimes lead to degradation.
> > > > > > > + Providing insights into synthesizing high-quality, complex mathematical QA pairs that could potentially benefit the entire community and inspire further research in this area.
> > > > > > >
> > > > > > > **In fact, our method is still effective even under low-cost settings with a small set of synthesized data.** Specifically, when fine-tuning with only 10-20% of our synthesized data, it achieves state-of-the-art performance on Llama3-8B, with significantly less data compared to other baseline methods (see the table below). Aside from the cost, our pipeline is highly user-friendly and requires only the ability to call APIs, making it easy to switch between different models for use. This simplicity ensures accessibility and ease of adoption.
> > > > > > >
> > > > > > > | Method    | Model     | Size(k) | GSM8K | MATH | College MATH | Olympiad | TabMWP | TheoremQA | AMC2023 | AIME2024 |
> > > > > > > |-----------|-----------|---|-------|------|--------------|----------|--------|-----------|---------|---------|
> > > > > > > | MetaMath  | Llama3-8B | 395 | 80.5  | 32.6 | 19.3         | 6.7      | 54.1   | 13.1      | 6/40    | 0/30    |
> > > > > > > | DART-Math | Llama3-8B | 585  | 81.8  | 46.9 | 28.4         | 15.9     | 66.3   | 20.5      | 8/40    | 1/30    |
> > > > > > > | MAmmoTH2  | Llama3-8B | 10000  | 69.6  | 33.4 | 32.3         | 8.1      | 43.8   | **29.7**      | 7/40    | 0/30    |
> > > > > > > | MathScale | Llama3-8B | 2021 | 70.8  | 34.6 | 22.5         | 9.0      | 74.3   | 18.9      | 2/40    | 1/30    |
> > > > > > > | _Wisdom_    | Llama3-8B | 244 | 83.7  | 55.6 | 35.0         | **23.6**     | 81.7   | 24.8      | 10/40   | 1/30    |
> > > > > > > | _Wisdom_    | Llama3-8B | 322 | **84.5**  | **57.4** | **36.7**         | 23.3     | **82.0**   | 28.5      | **12**/40   | **1**/30    |
> > > > > > >
> > > > > > > > Methodologically, perhaps the main novelty here is combining distillation with the synthesis of novel problems, getting reliability via various consistency methods.
> > > > > > > > (here, I believe the "expert teacher" model is still roughly an upper bound, which the current results support - I wouldn't expect the WISDOM student model to get beyond what the teacher is capable of, so that's why I'm understanding it as fundamentally a distillation approach).
> > > > > > >
> > > > > > > Thank you for your insightful interpretation of our method. We would like to expand on this. Unlike traditional teacher-student distillation, which leverages weak signals or latent space capabilities from the teacher model, our method focuses on data synthesis by utilizing the capabilities of stronger teacher models. **While this is indeed a form of distillation, synthesizing data in the natural language domain does not always work seamlessly, as it poses unique challenges compared to other forms of distillation.**
> > > > > > >
> > > > > > > As mentioned in the Introduction (lines 57–59), a lack of high-quality, high-difficulty, and high-diversity training data could lead to performance degradation of the student model. Similarly, if synthesized data does not emphasize difficulty and diversity, large-scale data synthesis may negatively impact the student model. This is why, even though our method performs well with a small amount of data, we scaled the synthesized dataset to millions of samples to ensure the robustness and effectiveness of our approach.
> > > > > > >
> > > > > > > Our main contribution lies not only in “the synthesis of novel problems” but also in effectively enhancing the difficulty and diversity of problems. This ensures that our approach remains valid even when scaling to millions of samples, while still achieving efficient improvements in the complex mathematical reasoning abilities of the student model with smaller datasets.
> > > > > > >
> > > > > > > We appreciate and acknowledge your understanding of our approach. Although the “expert teacher” model indeed represents an upper bound for the WISDOM student model, the synthesized data can be combined with other methods (e.g., reinforcement learning or additional open-source datasets) to further push the boundaries of the student model’s performance.
> > > > > > >
> > > > > > >
> > > > > > > > Algorithm line 27 should have an union
> > > > > > >
> > > > > > > Thank you for pointing that out. We will correct it in the revision.
> > > > > > >
> > > > > > > ---
> > > > > > >
> > > > > > > Overall, we are very grateful for your active participation in the discussion. Your recognition of our contributions, along with your insightful comments, has been invaluable to improving our work. Thank you again!

---

### Official Review · Reviewer_8YRZ · 2024-11-04

**Soundness:** 1
**Presentation:** 2
**Contribution:** 1
**Rating:** 1
**Confidence:** 5

**Summary:**

WISDOM is a framework for improving the mathematical reasoning capabilities of Large Language Models (LLMs). The framework achieves this through progressive curriculum synthesis. It generates high-quality Chain-of-Thought (CoT) training data in increasing difficulty levels using a combination of weak and expert teachers. Specifically, it has three main stages: weak teacher guiding, critical expert teaching, and experts consistency voting. It also has a mechanism for evolving harder questions. The authors then use this data to create WISDOM series models. The authors evaluate WISDOM on multiple mathematical reasoning benchmarks and show improvements over existing approaches, particularly for smaller models.

**Strengths:**

This is an interesting paper with some key strengths. The progressive synthesis approach, moving from easy to difficult problems through multiple stages, represents an advancement over existing methods that often treat all problems with equal weight or rely heavily on majority voting.

The authors show that even smaller models (7B-8B parameters) can achieve competitive performance with much larger models on certain datasets when trained using their WISDOM framework. The paper's has thorough ablation studies and careful analysis of different components, including the embeddings. The authors provide detailed investigations into the impact of knowledge base integration, answer consistency, and scaling effects. The authors demonstrate strong performance on challenging out-of-domain problems like AIME2024 and AMC2023. I appreciated the authors' careful attention to data contamination prevention using the 10-gram hash deduplication method.

In terms of clarity, I do appreciate the variety of visuals and the algorithm, but I have many clarity concerns in the weaknesses below.

**Weaknesses:**

I think there are quite a few areas that I am unclear on or see some overclaiming in.

1. I agree that there is a lack of data in this field. However, I think you need to back up statements like these with citations: “However, open-source datasets only contain a relatively low proportion of high-quality data.” You also have to cite other topics you mention, like in this sentence: “Although the widely adopted Rejection Sampling can generate data without reducing the difficulty level of the instructions, the reliance on ground truth limits its broader applicability.”
2. In Table 2, I recommend that the authors clearly label the y-axis and specify which benchmark(s) the accuracy values correspond to in the figure caption or main text.
3. It should be clear upfront how you measure problem difficulty. I recommend adding a brief explanation in the introduction, potentially with some context of curriculum learning, of how problem difficulty is measured or assessed in your approach.
4. There are many sentences such as that that are very qualitative but not quantitative: “Following curriculum learning principles, we first employ a weak but cost-effective teacher to solve a large number of easy problems. Subsequently, a strong but more resource-intensive expert is used for medium-difficulty questions, thereby optimizing resource utilization.” I am left wondering what medium-difficulty questions mean and what counts as resource-intensive.
5. Overall, I find the methodology section to be confusing. The difficulty progression mechanism needs more detailed exposition. Can you specify how difficulty is specifically measured or controlled across the three stages so I can better understand the progression from "easy" to "hard"? Moreover, the relationship between problem types and difficulty progression requires clarification. It would be valuable to understand whether certain mathematical domains follow different difficulty progression patterns.
6. Additionally, this paper lacks an explanation of the real world use case of WISDOM. Is the expectation that the user use WISDOM to generate data for their own base models? Isn’t this expensive for mathematicians? Or are mathematicians not the target audience?
7. Can you please define the abbreviation "GT-Free" in the table caption or in the text discussing Table 1?
8. What do you mean it “surpasses the 60% threshold for the first time”? Don’t all of the methods in the first 4 rows do the same?
9. It seems that most of the improvement is in AMC2023 and AIME2024. Although you claim SOTA in other benchmarks like College MATH, I wouldn’t say the improvements are “outstanding”
10. Given that Critical Expert Teaching only has a small improvement, can you please justify the inclusion of the Critical Expert Teaching stage? It seems that a big part of your claims is your focus on computational speedup and cost efficiency, so I would like to understand the aspects of this stage that outweigh the computational costs.
11. I don’t see an analysis of “progressing” over time. It seems that you just present at the end of the progressive synthesis.
12. I would also like to see more theoretical analysis of why this works, specifically with reference to theorem proving as as field.
13. Is this really curriculum learning if most advancements are only in the hardest theorems?


Small writing details:
I think the title should say “Mathematical Reasoners” (plural)
“prior arts in mathematical data synthesis” should likely be “prior works in mathematical data synthesis”

**Questions:**

I already have some questions in weaknesses above, but here are some more:

1. How are you measuring difficulty?
2. What are the computational requirements for implementing WISDOM in practice?
3. The paper claims cost efficiency compared to majority voting, but how does the total computational cost compare to other approaches in the literature?
4. Could you provide more analysis of the failure cases? Understanding where the approach doesn't work well would be valuable.
5. How sensitive is the approach to the choice of weak and expert teachers? What happens if you use different models in these roles?
6. How can users customize the curriculum for their specific needs?
7. Why haven’t you tried this approach with formal mathematics?
8. How have you verified that the answers or proofs are correct?

---

> ### Author Response · Authors · 2024-11-17
>
> # Response to Reviewer 8YRZ
>
> We sincerely thank the reviewers for their thorough reading of our paper and for raising detailed and insightful questions.
> We deeply appreciate the time and effort spent in providing constructive feedback, which has helped us improve the clarity and quality of our submission.
> Below, we address the key concerns and questions raised one by one.
> > W1: Some statements lack proper citations.
>
> **Response**: Thank you for the suggestions. **We will add the following references to support the statement**: "However, open-source datasets only contain a relatively low proportion of high-quality data."[1-6]
> Regarding the statement: "Although the widely adopted Rejection Sampling can generate data without reducing the difficulty level of the instructions, the reliance on ground truth limits its broader applicability", we believe no additional references are necessary due to the inherent limitations of Rejection Sampling (RS).
> RS is primarily designed to generate diverse answers rather than modifying instructions. As a result, it cannot enhance instruction difficulty and, in certain cases, may even reduce it. Furthermore, when newly synthesized problems lack explicit ground truth, RS fails to effectively guide the generation and selection of questions.
> These limitations are intrinsic to its design and sufficiently explain why no citations are needed.
>
> > W2: In Table 2, I recommend that the authors clearly label the y-axis and specify which benchmark(s) the accuracy values correspond to in the figure caption or main text.
>
> **Response**: Thank you for the feedback. We will consider incorporating the updates in the final revision.
>
> > W3,W5 and Q1: (1) It should be clear upfront how you measure problem difficulty. I recommend adding a brief explanation in the introduction, potentially with some context of curriculum learning, of how problem difficulty is measured or assessed in your approach;
> > (2) Overall, I find the methodology section to be confusing. The difficulty progression mechanism needs more detailed exposition. Can you specify how difficulty is specifically measured or controlled across the three stages so I can better understand the progression from "easy" to "hard"? Moreover, the relationship between problem types and difficulty progression requires clarification. It would be valuable to understand whether certain mathematical domains follow different difficulty progression patterns;
> > (3) How are you measuring difficulty?
>
> **Response**: We sincerely thank the reviewer for the constructive comments. In this paper, **we measure the difficulty with the response consistency of LLM**. The motivation comes from an intuitive hypothesis: simpler problems are more likely to yield consistent results across various solutions, including CoT and PoT.
>
> To verify this, **we design experiments to explore the relationship between inner response consistency and problem difficulty**.
> Using a specific prompt (see Appendix B.4) and the DeepSeek-V2.5 model (note: the V2.0 API has been deprecated due to updates), we test the relationship between response consistency and difficulty on 5,000 test problems from the MATH dataset.
> Difficulty is measured using the “Level” tags provided by the dataset.
> **The results (see the below table,revision in Appendix Table 9) show a clear trend: as problem difficulty increases, response consistency decreases. This trend strongly suggests that response consistency is a simple yet effective metric for evaluating problem difficulty. Furthermore, as shown in Tables 6 and 8, we observe that response consistency positively contributes to improving mathematical reasoning performance, both in terms of accuracy and quality**.
>
> |                           | Level 1 | Level 2 | Level 3 | Level 4 | Level 5 |
> |---------------------------|---------|---------|---------|---------|---------|
> | Response Consistency Rate | 75.3    | 70.6    | 65.0    | 62.1    | 54.2    |

---

> > ### Author Response · Authors · 2024-11-17
> >
> > > W4: There are many sentences such as that that are very qualitative but not quantitative: “Following curriculum learning principles, we first employ a weak but cost-effective teacher to solve a large number of easy problems. Subsequently, a strong but more resource-intensive expert is used for medium-difficulty questions, thereby optimizing resource utilization.” I am left wondering what medium-difficulty questions mean and what counts as resource-intensive.
> >
> > **Response**: Overall, the curriculum learning framework employs a progressive approach to increase the difficulty of problems through **weak model internal consistency, consistency between weak and strong models, and strong model consistency**, using a funnel-like filtering process. Medium-difficulty refers to problems that satisfy the consistency requirement between weak and strong models. Resource-intensive, on the other hand, indicates scenarios where even simple problems require multiple API calls to ensure response accuracy, thereby consuming significant resources.
> >
> > > W6: Additionally, this paper lacks an explanation of the real world use case of WISDOM. Is the expectation that the user use WISDOM to generate data for their own base models? Isn’t this expensive for mathematicians? Or are mathematicians not the target audience?
> >
> > **Response**: Our method is the first to simultaneously enhance data diversity and difficulty while fully accounting for cost efficiency, making it particularly suitable for real-world applications. Compared to other approaches, we optimize the data synthesis process to strike a balance between efficiency and effectiveness, minimizing computational and resource costs wherever possible.
> >
> > The WISDOM data synthesis method requires only basic API-calling capabilities in mathematics to generate high-quality data. This approach is simple to implement, cost-effective, and significantly lowers the technical barrier for generating complex data, thereby offering notable advantages for practical applications.
> >
> > Our method is model-agnostic and, in theory, can be adapted to any scenario where response consistency can be applied for data selection and synthesis. Furthermore, it is not limited to mathematical contexts; this framework can, in principle, be employed for any LLM learning task.
> > > W7: Can you please define the abbreviation "GT-Free" in the table caption or in the text discussing Table 1?
> >
> > **Response**: “GT-Free” refers to Ground Truth Free, indicating that, unlike methods such as Rejection Sampling and expert iteration, it does not require ground truth assistance when synthesizing new problems.
> >
> > > W8: What do you mean it “surpasses the 60% threshold for the first time”? Don’t all of the methods in the first 4 rows do the same?
> >
> > **Response**: Since our paper was completed during the summer, the statement “surpasses the 60% threshold for the first time” refers specifically to the context of open-source models available at that time. As for the “first 4 rows” these models are closed-source and were therefore excluded from our comparisons. This exclusion is based on the principle that directly comparing open-source models with closed-source models is not fair.
> >
> > > W9 and W13:  (1) It seems that most of the improvement is in AMC2023 and AIME2024. Although you claim SOTA in other benchmarks like College MATH, I wouldn’t say the improvements are “outstanding”; (2) Is this really curriculum learning if most advancements are only in the hardest theorems?
> >
> > **Response**: For a detailed description of the datasets, please refer to Appendix A.3.
> > Specifically, the test sets AMC2023 and AIME2024 consist of only 30 and 40 samples, respectively.  In contrast, our method demonstrates significant relative improvements over other SoTA models on larger datasets, such as College MATH(relatively **30.65%** in llama3-8B). This is primarily because the OOD (out-of-distribution) benchmark datasets we selected are inherently more challenging, making absolute score improvements more difficult to achieve.
> > Nevertheless, our method still exhibits notable performance gains across multiple datasets.

---

> > > ### Author Response · Authors · 2024-11-17
> > >
> > > > W10: Given that Critical Expert Teaching only has a small improvement, can you please justify the inclusion of the Critical Expert Teaching stage? It seems that a big part of your claims is your focus on computational speedup and cost efficiency, so I would like to understand the aspects of this stage that outweigh the computational costs.
> > >
> > > **Response**: Our core argument is not centered on computational acceleration or cost efficiency but rather on the evolution of instruction difficulty achieved through response consistency. The proposed Critical Expert Teaching method leverages not only the strong model’s inner response consistency but also incorporates the weak model’s response consistency, enabling more effective difficulty control and gradual instruction upgrading. This stage is crucial to the overall design. **If this stage is omitted, the large capability gap between the strong and weak models could result in problems deemed challenging for the weak model being classified as simple by the strong model. These problems would then bypass filtering and directly enter the final stage, consuming substantial computational resources and leading to inefficiencies.**
> > > By introducing consistency evaluations between the strong and weak models, we can filter out medium-difficulty problems, effectively avoid resource waste, and ensure both the precision and efficiency of instruction difficulty evolution.
> > >
> > > > W11: I don’t see an analysis of “progressing” over time. It seems that you just present at the end of the progressive synthesis.
> > >
> > > **Response**: For a detailed analysis, please refer to Sections 2.1 and 2.2, as well as Algorithm 1. Our method achieves progressive development of data not only across different stages but also through iterative rounds, where data scale is expanded, and data difficulty is further elevated.
> > >
> > > > W12: I would also like to see more theoretical analysis of why this works, specifically with reference to theorem proving as as field.
> > >
> > > **Response**: In this work, we primarily focus on the empirical study of data synthesis. The theoretical analysis of why it works, such as theorem proving, will be considered in future work.
> > >
> > > > Q2: What are the computational requirements for implementing WISDOM in practice?
> > >
> > > **Response**: WISDOM can be implemented using APIs provided by many closed-source large models, which significantly reduces training resource requirements and is cost-effective.
> > > Alternatively, for those with sufficient computational resources, the process can also be implemented by deploying open-source models on GPUs.
> > >
> > > > Q3: The paper claims cost efficiency compared to majority voting, but how does the total computational cost compare to other approaches in the literature?
> > >
> > > **Response**: Other methods either only support local deployment or fail to address the costs associated with calling GPT-4. In contrast, **we are the first to systematically consider real-world costs in our research and optimize for efficiency**. This design makes our approach more aligned with practical application needs and significantly enhances its practicality.
> > >
> > > > Q4: Could you provide more analysis of the failure cases? Understanding where the approach doesn't work well would be valuable.
> > >
> > > **Response**: We have added a detailed analysis of failure cases in Appendix. The failures in inner consistency are primarily caused by the following factors:
> > > 1.	**Precision Issues in Decimal Calculations (Appendix D.9) :**
> > > Large Language Models (LLMs) often struggle with achieving precision in decimal-related reasoning, leading to inconsistencies between Chain of Thought (CoT) and Program of Thought (PoT) results.
> > > 2.	**Decision Boundary Issues in Complex Algebra Problems (Appendix D.7-8) :**
> > > Certain algebra problems are highly complex. While PoT methods can effectively solve these problems by leveraging libraries such as Scipy and Math, CoT methods are limited by the decision boundaries of the model itself, making them less effective.
> > > 3.	**Excessive Problem Difficulty:**
> > > For extremely challenging problems, even strong models (e.g., GPT-4) fail to produce consistent outputs despite multiple attempts.

---

> > > > ### Author Response · Authors · 2024-11-17
> > > >
> > > > > Q5: How sensitive is the approach to the choice of weak and expert teachers? What happens if you use different models in these roles?
> > > >
> > > > **Response**: When both the Weak Teachers and Expert Teachers are relatively weak, synthesizing hard questions requires more iterative rounds, and the upper limit of question difficulty remains lower. Conversely, when the Weak Teachers are weak but the Expert Teachers are strong, fewer rounds are needed to synthesize hard questions, and the difficulty ceiling is higher. However, this significantly increases the burden on the Expert Teachers, leading to higher computational and resource costs.
> > > > Therefore, the configuration of teacher capabilities directly impacts the balance between efficiency and cost in question synthesis.
> > > >
> > > > > Q6: How can users customize the curriculum for their specific needs?
> > > >
> > > > **Response**: The flexible usage of WISDOM can be achieved simply by calling APIs, enabling customized configurations based on specific costs and question requirements.
> > > > +	When the budget is ample: Both high-performance Weak and Expert models can be utilized simultaneously, with different strong models selected to push the boundaries of the data, thereby generating higher-quality questions and responses.
> > > > +	When the budget is limited: A combination of open-source Weak models and relatively strong closed-source models can be employed. Through multi-round iterations, the question difficulty and response quality can be improved as much as possible.
> > > >
> > > > > Q7: Why haven’t you tried this approach with formal mathematics?
> > > >
> > > > **Response**: This pertains to another area of mathematics and is not the core focus of our work, we will implement this in the future work.
> > > >
> > > > > Q8: How have you verified that the answers or proofs are correct?
> > > >
> > > > **Response**: **In the field of data synthesis, newly generated problems often lack ground truth unless verified manually on a case-by-case basis.**
> > > > However, for newly generated problems at the scale of millions, manual verification is clearly impractical. Therefore, our goal during the synthesis process is to maximize response accuracy rather than ensure that every generated problem’s response is entirely correct. If verifying the correctness of generated responses is necessary, one can refer to the consistency rate between the generated responses and those from a strong model (response consistency), a method highlighted in relevant research from [7].
> > > > While correct responses undoubtedly enhance the effectiveness of large language models (LLMs) more significantly, the model’s performance can still improve even when some responses contain errors. This is because erroneous responses can increase data diversity to some extent, and they often include partially correct reasoning paths, providing additional learning signals for the model.
> > > >
> > > > > Small writing details: I think the title should say “Mathematical Reasoners” (plural) “prior arts in mathematical data synthesis” should likely be “prior works in mathematical data synthesis”
> > > >
> > > > **Response**: We appreciate this suggestion and have implemented the revision accordingly in the updated version.
> > > >
> > > > [1] What Makes Good Data for Alignment? A Comprehensive Study of Automatic Data Selection in Instruction Tuning. ICLR 2024
> > > > [2] Get More for Less: Principled Data Selection for Warming Up Fine-Tuning in LLMs. ICLR 2024
> > > > [3] AlpaGasus: Training a Better Alpaca with Fewer Data. ICLR 2024
> > > > [4] LESS: Selecting Influential Data for Targeted Instruction Tuning. ICML 2024
> > > > [5] MAmmoTH2: Scaling Instructions from the Web. CoRR abs/2405.03548 (2024)
> > > > [6] From Quantity to Quality: Boosting LLM Performance with Self-Guided Data Selection for Instruction Tuning. NAACL-HLT 2024
> > > > [7] Neuro-Symbolic Data Generation for Math Reasoning. NeurIPS 2024.

---

> > > > > ### Author Response · Authors · 2024-11-25
> > > > >
> > > > > Dear Reviewer 8YRZ,
> > > > >
> > > > > We hope this message finds you well. With the discussion period concluding in two days, we want to kindly follow up to see if our responses have addressed your concerns. We have carefully addressed each of the points you raised and provide additional experiments to support our arguments.
> > > > >
> > > > > We greatly value your feedback and are eager to engage in further discussion if there are any remaining questions or comments. Please let us know if there’s anything else we can clarify—we’d be more than happy to respond.
> > > > >
> > > > > Thank you once again for your time and constructive insights.
> > > > >
> > > > > Best regards,
> > > > >
> > > > > The Authors

---

> > > > > > ### Author Response · Authors · 2024-11-29
> > > > > > **A gentle reminder**
> > > > > >
> > > > > > Dear Reviewer 8YRZ,
> > > > > >
> > > > > > We would like to kindly remind you about the rebuttal we submitted in response to your valuable feedback. **We would greatly appreciate it if you could take a moment to review our responses**.
> > > > > >
> > > > > > We are eager to know if our rebuttal has sufficiently clarified your concerns, or if there are any additional points you would like to discuss further.
> > > > > >
> > > > > > Thank you once again for your time and thoughtful consideration.
> > > > > >
> > > > > > Sincerely,
> > > > > >
> > > > > > The Authors

---

> > > > > > > ### Comment · Reviewer_8YRZ · 2024-12-02
> > > > > > > **Response 1**
> > > > > > >
> > > > > > > I appreciate the authors’ detailed response to my concerns. However, I find the author response insufficient. You committed to addressing my comment on labeling the y-axis and specifying benchmarks in Table 2, yet this remains unaddressed in the revision. Clear labeling and contextualization are essential for accurate interpretation of your results.
> > > > > > >
> > > > > > > The response you provided about methodology does not adequately address my concerns. Your response does not explain how the progression mechanism (from "easy" to "hard") is controlled or implemented across the three stages. While you state that response consistency is used to measure difficulty, you do not provide a detailed description of how this metric influences progression in practice, particularly within the stages of Weak Teacher Guiding, Critical Expert Teaching, and Experts Consistency Voting. What thresholds or criteria are used to move problems between these stages? You still have not addressed whether difficulty progression differs across mathematical domains. Are there any variations in how different domains or types of problems behave with respect to difficulty progression? This remains an unexplored dimension that could significantly enhance the methodological clarity and impact of your work. Your response does not acknowledge or address the broader critique that the methodology section is confusing. A clearer, structured explanation of the progression mechanism and its relation to response consistency is essential.
> > > > > > >
> > > > > > > The explanation of "medium-difficulty" as problems satisfying consistency between weak and strong models is qualitative and does not define how this consistency is measured. Are there thresholds or specific metrics that classify problems as "medium-difficulty"? This remains ambiguous. Similarly, the description of "resource-intensive" as involving "multiple API calls" does not quantify the extent of resource usage. How many calls, on average, qualify as resource-intensive? Is there a measurable cost or runtime threshold? The authors describe the use of a "funnel-like filtering process," but there are no clear explanations of how this process operates in practice. How are problems selected or filtered at each stage? What specific criteria determine progression or rejection? The response does not explain what distinguishes problems of "medium-difficulty" from "hard" problems in terms of model behavior or outcomes. My original concern was the qualitative nature of such statements, yet the response repeats the same style of reasoning without providing concrete metrics or examples. For instance, what percentage of problems typically fall into the "medium-difficulty" category? How does resource utilization compare across the three stages?
> > > > > > >
> > > > > > > The authors' response does not sufficiently address the question of real-world use cases and target audience. It provides general statements about the method's flexibility and cost-effectiveness but fails to engage with the specific concerns raised. The response avoids directly addressing whether WISDOM is intended for mathematicians, general LLM researchers, or other specific user groups. My concern about the accessibility and utility for mathematicians is left unanswered. The authors claim that WISDOM is "cost-effective" and "lowers the technical barrier," but they do not provide concrete evidence or examples to substantiate these claims. For example, what is the estimated cost of generating data using WISDOM for a typical use case? How does it compare to existing methods? The authors state that WISDOM is model-agnostic and adaptable, but this does not address the core question of how users are expected to employ WISDOM in practice. Are mathematicians or researchers expected to invest significant resources into API calls, or will pre-generated datasets be made available?
> > > > > > >
> > > > > > > The authors' response about the 60% threshold fails to adequately justify their claim and creates unnecessary confusion. The phrase “surpasses the 60% threshold for the first time” is misleading because it does not clarify that this milestone is specific to open-source models. Readers are likely to interpret the statement more broadly, as encompassing all methods listed in the table. The authors should have explicitly stated this distinction upfront. While the authors argue that closed-source models were excluded from comparisons for fairness, this context is missing in the paper. If closed-source models are excluded from certain claims, it should be explicitly stated wherever relevant, particularly when highlighting "firsts" or milestones. While it is valid to avoid direct comparisons between open-source and closed-source models for fairness, the authors do not explain why this principle applies in this specific instance. For example, why is it unfair to note that GPT-4o and similar models also surpassed the 60% threshold, even if they are closed-source? This omission weakens the justification for their exclusion.

---

> > > > > > > ### Comment · Reviewer_8YRZ · 2024-12-02
> > > > > > > **Response 2**
> > > > > > >
> > > > > > > It is unclear if the relative improvement of 30.65% on College MATH compared to llama3-8B should be described as "outstanding,” especially since all closed source models outperform this method. The response does not address whether the observed improvements align with the principles of curriculum learning. If most advancements are concentrated on the hardest problems, it raises questions about whether the method genuinely facilitates progressive learning or simply benefits from solving challenging problems better than existing approaches. The response does not elaborate on the "notable performance gains" across other datasets. This again feels like overclaiming.
> > > > > > >
> > > > > > > The authors’ response provides some justification for the Critical Expert Teaching stage but leaves critical gaps in addressing the original concern. While the authors claim their argument is centered on instruction difficulty evolution rather than computational efficiency, the paper does emphasize cost efficiency in several places. This inconsistency weakens their argument. If computational efficiency is not a focus, why is it mentioned as a key benefit in the paper? The authors assert that the Critical Expert Teaching stage is crucial for avoiding inefficiencies but provide no empirical evidence to demonstrate how this stage meaningfully contributes to cost savings or performance improvements. While the authors discuss avoiding inefficiencies, they do not provide a clear cost-benefit analysis of including this stage.
> > > > > > >
> > > > > > > The authors' response does not adequately address the critique and leaves important gaps in explaining how "progressing" over time is analyzed. Referring to Sections 2.1, 2.2, and Algorithm 1 does not address the core concern. While these sections describe the process of data synthesis, they do not explicitly analyze or present evidence of "progression" in terms of measurable improvements over time. This oversight fails to support the claim of progressive synthesis. The paper seems to focus on the final outcomes of the synthesis process, but the critique highlights the absence of insights into intermediate steps. For example, how does the performance improve across rounds or stages? Is there evidence that difficulty evolves consistently as intended? The authors assert that data difficulty is "further elevated" across iterative rounds, but they do not detail how this elevation is measured or validated. Without clear metrics or visualizations (e.g., difficulty progression graphs), this statement lacks credibility.
> > > > > > >
> > > > > > > The response dismisses a valid request for theoretical grounding without offering even a preliminary effort to address it. While the paper focuses on empirical results, theoretical analysis is critical for providing deeper insights into why the method works. Simply deferring this to future work without any attempt to discuss potential theoretical underpinnings weakens the impact of the paper.
> > > > > > >
> > > > > > > The response does not specify the computational resources required for either option. For example, what kind of hardware (e.g., GPU specifications) is needed for deploying open-source models? What are the approximate API usage costs for implementing WISDOM via closed-source models? While the response mentions that API-based implementation is "cost-effective," no cost estimates or comparisons with alternative methods are provided to substantiate this claim.
> > > > > > >
> > > > > > > The response fails to quantify how WISDOM's computational costs compare to other approaches in the literature. Without numerical evidence or benchmarks, the claim of cost efficiency lacks credibility. There is no mention of specific cost metrics (e.g., time, computational resources, or monetary cost) for WISDOM relative to other methods. A proper comparison would include figures such as cost per example generated or total cost for a typical dataset. How does the "funnel-like" filtering mechanism reduce computational costs? How does it scale with dataset size?
> > > > > > >
> > > > > > > The authors' response on failure modes is a step in the right direction, but it lacks sufficient depth and actionable insights. While the authors identify three categories of failure cases, the explanation for each is too high-level. How often do these failures occur (e.g., as a percentage of total cases)? The response does not discuss how these failure cases could be mitigated or addressed in future iterations of the method. Understanding limitations is valuable, but identifying potential solutions is just as important. Are certain datasets or problem types disproportionately affected? How do these failures influence the accuracy or efficiency metrics?

---

> > > > > > > ### Comment · Reviewer_8YRZ · 2024-12-02
> > > > > > > **Response 3**
> > > > > > >
> > > > > > > The response does not provide concrete data or metrics to substantiate claims about the impact of teacher choice. How many additional rounds are required when both teachers are weak? How much does the computational cost increase when using stronger expert teachers? There is no mention of experiments conducted to test the sensitivity to teacher choice. Such results would lend credibility to the author claims. The response does not offer practical insights into how to optimize the choice of teachers for different use cases or datasets. For example, should users prioritize stronger experts if computational resources are limited?
> > > > > > >
> > > > > > > The response gives general advice on budget-based configurations but does not delve into how users can customize the curriculum structure itself (e.g., difficulty progression, problem domains, or the number of iterations). The response does not specify what aspects of WISDOM are customizable (e.g., thresholds for problem difficulty, the number of problems generated at each stage, or the criteria for progression between stages). The response does not provide concrete examples of how users might tailor the method for specific use cases, such as different mathematical domains, education levels, or research goals.
> > > > > > >
> > > > > > > The response emphasizes response consistency but does not directly address how well this metric correlates with correctness. Consistency does not guarantee correctness, especially if both responses are systematically flawed. The response does not quantify how often generated responses are correct or how often response consistency aligns with correctness. Without this data, it’s difficult to evaluate the reliability of the method. While the authors argue that errors can provide diversity and learning signals, they fail to specify how error rates are controlled to avoid diminishing the quality of synthesized data. The response dismisses manual verification due to scale but does not mention any alternative strategies, such as sampling for quality checks or automated verification through external tools.
> > > > > > >
> > > > > > > I have decided to change my rating from 5 to 1 because the authors have consistently failed to adequately address critical feedback, both in their initial submission and in their response. The recurring issues with vague claims, lack of quantitative analysis, and superficial justifications undermine the credibility of the paper. Furthermore, key concerns about methodology, real-world applicability, and theoretical grounding remain unresolved. This lack of engagement with constructive critique and the absence of meaningful revisions indicate a fundamental disconnect between the authors’ responses and the expectations of rigorous academic standards at ICLR.

---

> > > > > > > > ### Author Response · Authors · 2024-12-03
> > > > > > > >
> > > > > > > > Regarding the numerous misleading aspects raised by the reviewer in the feedback, we chose to focus solely on addressing the **core misleading points**, given the proximity of the discussion deadline.
> > > > > > > >
> > > > > > > > > While you state that response consistency is used to measure difficulty, you do not provide a detailed description of how this metric influences progression in practice, particularly within the stages of Weak Teacher Guiding, Critical Expert Teaching, and Experts Consistency Voting. What thresholds or criteria are used to move problems between these stages? You still have not addressed whether difficulty progression differs across mathematical domains.
> > > > > > > > > The explanation of "medium-difficulty"
> > > > > > > >
> > > > > > > > In Section 2.2, we have **thoroughly analyzed** how question difficulty is progressively upgraded across the three stages using response consistency. Additionally, we validated the relationship between response consistency and question difficulty in Appendix Table 9. In our revision (lines 70–71), we **explicitly clarified** that our approach employs the model itself to implement curriculum learning, during which difficulty is progressively upgraded. This is not done without a performance threshold but is instead guided by model performance.
> > > > > > > >
> > > > > > > > The advantage of this method lies in its ability to generate relatively controllable difficulty levels based on the existing model. We **kindly request** the reviewer to carefully review our paper. Our experimental results fully support our hypothesis.
> > > > > > > > > The authors describe the use of a "funnel-like filtering process," but there are no clear explanations of how this process operates in practice. How are problems selected or filtered at each stage?
> > > > > > > >
> > > > > > > > Our use of response consistency to implement the “funnel-like filtering process” and difficulty progression is evident in **multiple sections** of the paper, including Algorithm 1 and Section 2. We kindly request the reviewer to carefully review these parts of our work for a clearer understanding.
> > > > > > > >
> > > > > > > > > The authors' response about the 60% threshold fails to adequately justify their claim and creates unnecessary confusion.
> > > > > > > >
> > > > > > > > Please carefully review the context of this statement (lines 325–327). The text **explicitly states** that, using a model fine-tuned based on DeepSeek, we surpassed the 60% benchmark on the MATH dataset for the first time, compared to the previously fine-tuned sota model, Dart-Math, on DeepSeek.
> > > > > > > >
> > > > > > > > First, this statement is unambiguous as written. Second, it is only fair to compare models of the same size, rather than directly comparing our results with closed-source models. **If there are any misunderstandings**, we encourage early and constructive discussions to clarify them, rather than continuing to interpret our work based on incorrect assumptions.
> > > > > > > >
> > > > > > > > > It is unclear if the relative improvement of 30.65% on College MATH compared to llama3-8B should be described as "outstanding”
> > > > > > > >
> > > > > > > > Our statement regarding “the relative improvement of 30.65% on College MATH” refers to the improvement achieved by fine-tuning on different synthetic datasets while using the same LLaMA3-8B base, compared to the Sota. It does not refer to an improvement relative to the LLaMA3-8B base model itself. We kindly request the reviewer to carefully review our paper and rebuttal to fully understand this clarification.
> > > > > > > >
> > > > > > > > > If most advancements are concentrated on the hardest problems, it raises questions about whether the method genuinely facilitates progressive learning or simply benefits from solving challenging problems better than existing approaches.
> > > > > > > >
> > > > > > > > Closed-source models enhance their performance from **multiple angles**, including model size, pretraining data, and reinforcement learning strategies. In contrast, our approach focuses solely on improving model performance during the data fine-tuning stage using synthetic data. Therefore, directly comparing our method with closed-source models is evidently unfair.
> > > > > > > >
> > > > > > > > Furthermore, our approach is not limited to challenging datasets—**it also achieves outstanding results on simpler datasets like GSM8K**, as shown in Table 2. Our primary motivation is to address the observed limitations of LLMs in solving complex mathematical reasoning tasks and to enhance their capabilities in this domain. This does not mean that our method is **only** effective on complex datasets. Instead, it is a versatile solution aimed at advancing LLMs’ reasoning abilities across different levels of task difficulty.

---

> > > > > > > > > ### Author Response · Authors · 2024-12-03
> > > > > > > > >
> > > > > > > > > > While the authors claim their argument is centered on instruction difficulty evolution rather than computational efficiency, the paper does emphasize cost efficiency in several places. This inconsistency weakens their argument. If computational efficiency is not a focus, why is it mentioned as a key benefit in the paper?
> > > > > > > > >
> > > > > > > > > Instruction difficulty and computational efficiency are both key contributions of our work, and they do not **conflict** with each other. We leverage response consistency to enhance instruction difficulty and utilize computational efficiency to improve the quality of responses to hard questions.
> > > > > > > > >
> > > > > > > > > It is clear that these two contributions are **fully complementary** and work together to holistically improve the quality of synthetic data. This synergy underscores the strength of our approach in advancing both the effectiveness and practicality of synthetic data for fine-tuning LLMs.
> > > > > > > > > > While these sections describe the process of data synthesis, they do not explicitly analyze or present evidence of "progression" in terms of measurable improvements over time. This oversight fails to support the claim of progressive synthesis. The paper seems to focus on the final outcomes of the synthesis process, but the critique highlights the absence of insights into intermediate steps
> > > > > > > > >
> > > > > > > > > We have **clearly explained** that our notion of progressing refers to the difficulty escalation and diversity of synthetic data through the “funnel-like filtering process.” In Table 3, we discussed the performance improvements across different stages, and we also provided additional performance gains across rounds (see the supplementary table below). These details were not included in the main paper because rounds, compared to stages, are less critical and can be adjusted flexibly based on specific needs.
> > > > > > > > >
> > > > > > > > > Moreover, we **have emphasized** that our approach is based on response consistency to achieve instruction difficulty evolution. While the difficulty of known labeled problems has been thoroughly validated, newly synthesized problems are novel and lack predefined difficulty labels. As such, their difficulty can **only** be measured using the model’s intrinsic response consistency. This methodology ensures that our approach is both effective and **adaptable to unseen data**.
> > > > > > > > >
> > > > > > > > > | Round            | Model     | Size(k) | GSM8K    | MATH     | College MATH | Olympiad | TabMWP   | TheoremQA | AMC2023   | AIME2024 |
> > > > > > > > > |------------------|-----------|---------|----------|----------|--------------|----------|----------|-----------|-----------|----------|
> > > > > > > > > | Round(1)         | Llama3-8B | 244     | 83.7     | 55.6     | 35.0         | 23.6     | 81.7     | 24.8      | 10/40     | 1/30     |
> > > > > > > > > | Round(1+2)       | Llama3-8B | 322     | **84.5** | 57.4     | 36.7         | 23.3     | 82.0     | 28.5      | 12/40     | 1/30     |
> > > > > > > > > | Round(1+2+3)     | Llama3-8B | 723     | 84.2     | 58.7     | 36.9         | 25.0     | 83.1     | 28.0      | 12/40     | 1/30     |
> > > > > > > > > | Round(1+2+3+4)   | Llama3-8B | 1050    | **84.5** | 59.2     | 38.9         | **26.1** | **84.0** | 28.4      | 13/40     | 0/30     |
> > > > > > > > > | Round(1+2+3+4+5) | Llama3-8B | 1467    | 83.2     | **59.7** | **42.2**     | 25.6     | 83.0     | **28.6**  | **17**/40 | **1**/30 |
> > > > > > > > >
> > > > > > > > > >  The authors assert that data difficulty is "further elevated" across iterative rounds, but they do not detail how this elevation is measured or validated.
> > > > > > > > >
> > > > > > > > > First, we would like to **clarify** that the increase in data difficulty is achieved through three functionally distinct stages, not through iterative rounds. Regarding the increase in difficulty, we have already explained in principle that the difficulty progressively increases across stages. This progression is achieved using a funnel-like approach that enhances instruction difficulty through Inner Consistency of Weak Models, Consistency Between Strong and Weak Models, and Inner Consistency of Strong Models.
> > > > > > > > >
> > > > > > > > > The data retained in later stages, having failed to achieve consistency in earlier stages, is **inherently more difficult** by principle. As demonstrated in Appendix Table 9, we have empirically shown a strong correlation between difficulty and consistency. Therefore, whether the data in the Experts Consistency Voting Stage is more difficult than the data from the Weak Teacher Guiding Stage does not necessarily require further graphical demonstration or quantification of difficulty.
> > > > > > > > >
> > > > > > > > > That said, if the reviewer is interested in observing the difficulty trend, we could sample a portion of the data and conduct consistency experiments for reference. However, such a request should have been raised **earlier** during the review process, rather than on the final day of the rebuttal period.

---

> > > > > > > > > > ### Author Response · Authors · 2024-12-03
> > > > > > > > > >
> > > > > > > > > > > The response dismisses a valid request for theoretical grounding without offering even a preliminary effort to address it.
> > > > > > > > > >
> > > > > > > > > > We would like to **reiterate**, in alignment with the discussions in other contemporary works [1-5], that this study primarily focuses on the empirical investigation of data synthesis.
> > > > > > > > > >
> > > > > > > > > > >  While the authors identify three categories of failure cases, the explanation for each is too high-level.
> > > > > > > > > >
> > > > > > > > > > We do not believe that our responses are high-level abstract and hard to understand. Instead, it is grounded in the **practical challenges** that LLMs commonly face when solving mathematical problems, where errors are frequent. A classic solution to this issue involves using **Python** programming to compute and verify results. However, such an approach **falls outside the scope** of our paper’s discussion.
> > > > > > > > > >
> > > > > > > > > > > The response does not provide concrete data or metrics to substantiate claims about the impact of teacher choice.
> > > > > > > > > >
> > > > > > > > > > We stand by our previous response to Q5. The specific usage of our method depends on individual budgets and expected outcomes. **This flexibility is a strength, not a weakness.** Our use of response consistency can be adapted or replaced based on specific needs, rather than being rigidly tied to a fixed pipeline. This adaptability enhances the practicality and versatility of our approach.
> > > > > > > > > >
> > > > > > > > > > > The response emphasizes response consistency but does not directly address how well this metric correlates with correctness.
> > > > > > > > > >
> > > > > > > > > >
> > > > > > > > > > In Section 3.6, we **thoroughly validated** the relationship between response consistency and accuracy from multiple perspectives. For newly synthesized problems that lack ground truth, we employed two approaches to verify this relationship:
> > > > > > > > > > 1.	**Comparison with GPT-4o**: Following the method adopted by [5], we compared the results with those of GPT-4o (currently one of the strongest models). Improvements in consistency with GPT-4o serve as evidence of improved accuracy.
> > > > > > > > > > 2.	**Fine-tuning with consistent vs. inconsistent data**: We fine-tuned the model using datasets with and without response consistency and evaluated the performance on both in-domain and out-of-domain benchmarks. Since high-quality, accurate responses naturally outperform incorrect ones, the results further support our approach (see experimental results in Appendix Table 8).
> > > > > > > > > >
> > > > > > > > > > These validations demonstrate the effectiveness of response consistency in enhancing data quality and improving model performance.
> > > > > > > > > >
> > > > > > > > > > > While the authors argue that errors can provide diversity and learning signals, they fail to specify how error rates are controlled to avoid diminishing the quality of synthesized data
> > > > > > > > > >
> > > > > > > > > > As **emphasized** throughout the main methodology of our paper, the design of Experts Consistency Voting with majority voting for challenging problems is specifically aimed at minimizing error rates and improving response quality. For newly synthesized data that lack ground truth, the strategies to control error rates have already been detailed in Section 3.6.
> > > > > > > > > >
> > > > > > > > > > Improving accuracy is simply another way of describing error rate control, and our approach ensures that newly generated data maintain high quality through these mechanisms.

---

> > > > > > > > > > > ### Author Response · Authors · 2024-12-03
> > > > > > > > > > >
> > > > > > > > > > > > I have decided to change my rating from 5 to 1 because the authors have consistently failed to adequately address critical feedback, both in their initial submission and in their response.
> > > > > > > > > > >
> > > > > > > > > > >
> > > > > > > > > > > Regarding the numerous issues raised previously, despite the fact that the reviewer appears to **lack a deep understanding** of both the LLMs domain and the field of synthetic data, and has demonstrated some misleading assumptions, we have **nonetheless provided** a thorough explanation of our approach and contributions, while **seeking constructive interaction throughout the review period**.
> > > > > > > > > > >
> > > > > > > > > > > However, despite multiple gentle reminders, the reviewer failed to engage actively. **On the final day of the rebuttal period**, the reviewer provided an **emotional review** with numerous misleading points (initially scoring the paper with an overall score of 5 and a confidence score of 3, which was later changed to an overall score of 1 and a confidence score of 5).
> > > > > > > > > > >
> > > > > > > > > > > We have **reasonable grounds to believe** that the reviewer, without a proper understanding of the role of synthetic data in mathematical reasoning for LLMs, did not adequately read or comprehend our paper and rebuttal response during much of the review period, instead providing an emotional response at the final deadline.
> > > > > > > > > > >
> > > > > > > > > > > Given the reviewer’s overall performance, we **have escalated the matter to the Area Chair**  and provided relevant details. We trust the AC will make a fair and impartial final decision.
> > > > > > > > > > >
> > > > > > > > > > > [1] DART-Math: Difficulty-Aware Rejection Tuning for Mathematical Problem-Solving. NeurIPS 2024.
> > > > > > > > > > >
> > > > > > > > > > > [2] METAMATH: BOOTSTRAP YOUR OWN MATHEMATICAL QUESTIONS FOR LARGE LANGUAGE MODELS. ICLR 2024.
> > > > > > > > > > >
> > > > > > > > > > > [3] MAMMOTH: BUILDING MATH GENERALIST MODELS THROUGH HYBRID INSTRUCTION TUNING. ICLR 2024.
> > > > > > > > > > >
> > > > > > > > > > > [4] MathScale: Scaling Instruction Tuning for Mathematical Reasoning. ICML 2024.
> > > > > > > > > > >
> > > > > > > > > > > [5] Neuro-Symbolic Data Generation for Math Reasoning. NeurIPS 2024.

---

### Official Review · Reviewer_GUQf · 2024-11-04

**Soundness:** 2
**Presentation:** 3
**Contribution:** 3
**Rating:** 6
**Confidence:** 4

**Summary:**

This study presents a data-centric research endeavor that progressively synthesizes high-quality CoT data from simple to complex. Empirical evidence substantiates the efficacy of the proposed approach.

**Strengths:**

1. The experiment was meticulously conducted with a wide array of comparative methods.
2. The article is well-written, presenting information in a coherent and easily comprehensible manner.

**Weaknesses:**

1. The proposed method lacks novelty, as it primarily represents a combination of several well-established techniques, including self-consistency, teacher model distillation, and the use of meta-data to guide data generation.

2. The discussion regarding the difficulty levels of the tasks is insufficient and does not adequately analyze relevant prior work [1].

3. Despite achieving high performance, the proposed data-centric approach faces a fundamental issue: the comparisons made are not entirely fair due to variations in the scale of data used and the differing levels of used prior knowledge.

4. The manuscript does not focus on a clear scientific problem; rather, it resembles an engineering-focused endeavor and lacks the rigor typically associated with scientific inquiry.

[1] Neuro-Symbolic Data Generation for Math Reasoning. NeurIPS 2024.

**Questions:**

1. How does this paper define easy and hard math problems?
2. Why does the author believe that adding more "easy to hard" data can address complex reasoning, or is it merely a matter of data fitting?

---

> ### Author Response · Authors · 2024-11-17
>
> # Response to Reviewer GUQf
>
> We sincerely appreciate the reviewers for the time and effort in evaluating our paper and for offering valuable and constructive feedback.
> Your insights and questions have been instrumental in helping us refine and strengthen our work. Below, we address each of your comments and concerns in detail, striving to provide clear explanations and additional clarifications wherever needed.
> We are more than willing to engage in further discussions to ensure that our contributions are conveyed with the utmost clarity and precision.
> > W1: The proposed method lacks novelty, as it primarily represents a combination of several well-established techniques, including self-consistency, teacher model distillation, and the use of meta-data to guide data generation.
>
> **Response**: Firstly, our proposed method WISDOM, is not a simple combination of existing modules, instead, it is carefully designed inspired by human curriculum learning. Humans often solve complex reasoning problems by breaking them down into smaller, solvable sub-problems, and learn to solve the question from easy to hard.
> Meanwhile, current data synthesis works, such as MetaMath, MAmmoTH2, NuminaMath, MathScale, **focus on generating diverse problems with limited effort on the difficulty of instructions**. And other works, such as DART-Math consider the difficulty-aware instruction in GSM8K, MATH for response synthesis, **which rely on ground truth for rejection sampling and there are no newly synthesized instructions.**
> Inspired by the curriculum learning of human, where individuals tackle complex reasoning problems by breaking them down into smaller, solvable sub-problems, which heavily relies on the accumulation and utilization of resolved fundamental problems, we propose WISDOM: progressively synthesizing complex reasoning questions from simpler ones to enhance the performance of LLMs on complex reasoning tasks, such as mathematical tasks, which effectively balances data diversity, breadth, diffculty, and computation cost.
>
> Secondly, it is worth noting that we refine and filter simpler problems based on **response consistency** without groundtruth, since the ground truth of synthesized questions cannot be obtained. The motivation comes from an intuitive hypothesis: simpler problems are more likely to yield consistent results across various solutions, including CoT and PoT. To verify this, we design experiments to explore the relationship between inner response consistency and problem difficulty. Using a specific prompt (see Appendix B.4) and the DeepSeek-V2.5 model (note: the V2.0 API has been deprecated due to updates), we test the relationship between response consistency and difficulty on 5,000 test problems from the MATH dataset. Difficulty is measured using the “Level” tags provided by the dataset. The results (see the below table and Appendix in Table 9) show a clear trend: as problem difficulty increases, response consistency decreases. This trend strongly suggests that response consistency is a simple yet effective metric for evaluating problem difficulty. Furthermore, as shown in Tables 6 and 8, we observe that response consistency positively contributes to improving mathematical reasoning performance, both in terms of accuracy and quality.
>
> |                           | Level 1 | Level 2 | Level 3 | Level 4 | Level 5 |
> |---------------------------|---------|---------|---------|---------|---------|
> | Response Consistency Rate | 75.3    | 70.6    | 65.0    | 62.1    | 54.2    |
>
> Overall, our curriculum learning framework WISDOM gradually increases problem difficulty through the following steps:
>
> 1.	**Inner consistency of weak models**: used to filter and refine simpler problems.
> 2.	**Consistency between strong and weak models**: used to filter medium-difficulty problems.
> 3.	**Inner consistency of strong models**: used to generate and optimize high-difficulty problems.
>
> During this process, we adopt a dynamic “funnel-like” filtering mechanism. Problems are not generated all at once; instead, seed data is iteratively updated over multiple rounds to synthesize new problems. Each module in our framework is indispensable. Without the inner consistency module for weak models, it would be impossible to filter simple problems in the most efficient manner. Without the consistency module between strong and weak models, problems that appear difficult for a weak teacher but are considered simple by a strong model (due to a significant performance gap between the weak teacher and the expert teacher) might bypass proper filtering. These problems would directly flow into the final stage, consuming substantial computational resources and leading to inefficiencies. Similarly, without the inner consistency module for strong models, it becomes challenging to ensure the quality of responses for challenging problems.

---

> > ### Author Response · Authors · 2024-11-17
> >
> > # Continue with the previous W1.
> > Ensuring the response quality for challenging problems is critical because we have observed that for high-difficulty tasks, solely relying on existing methods for data generation remains limited. This highlights the necessity of a robust mechanism to address these gaps and improve the overall efficiency and effectiveness of our approach. For example, we observe that GPT-4o-0513 achieves only a 6.7% success rate with greedy decoding on the AIME2024 dataset. Traditional methods often rely on single-pass greedy generation with GPT-4 for higher-difficulty problems, neglecting further improvements in response quality. We address this issue by conducting experiments to verify the importance of ensuring response quality in high-difficulty problems. Specifically, we compare retaining responses directly from single-pass GPT-4 greedy generation against using consistency-filtered results during the Experts Consistency Voting stage, keeping the dataset size constant. The results (see the below table) show that the former significantly degrades model performance. This highlights that improving response quality is a crucial factor when synthesizing high-difficulty problems.
> >
> > | Data          | Model     | S3 consistency | GSM8K | MATH | College MATH | Olympiad | TabMWP | TheoremQA | AMC2023 | AIME2024 |
> > |---------------|-----------|----------------|-------|------|--------------|----------|--------|-----------|---------|----------|
> > | Seed+S1+S2+S3 | DSMath-7B | ✖              | 79.3  | 58.3 | 37.8         | 26.1     | 83.4   | 33.0      | 10/40   | 0/30     |
> > | Seed+S1+S2+S3 | DSMath-7B | ✔              | **83.3**  | **62.4** | **45.0**         | **28.9**     | **85.7**   | **34.9**      | **11**/40   | **2**/30     |
> >
> >
> > > W2: The discussion regarding the difficulty levels of the tasks is insufficient and does not adequately analyze relevant prior work [1].
> >
> > **Response**: As discussed in Weakness 1, we employ a progressive approach to increase problem difficulty through the inner consistency of weak models, the consistency between strong and weak models, and the consistency of strong models. We have also supplemented our experiments to demonstrate the relationship between response consistency and difficulty.
> >
> > Regarding [1], the authors propose a neuro-symbolic pipeline that utilizes auto-formalization techniques to generate mathematical problems represented in domain-specific languages. These problems are then transformed and translated back into natural language. Similar to our work, [1] focuses on increasing both problem diversity and difficulty; however, the implementation strategies differ significantly.
> > + **In terms of diversity enhancement**, the abstraction dimensions differ: our approach expands diversity within the natural semantic space using a knowledge base to assist metadata, while [1] leverages a formula-based approach to project problem attributes into the formal proof space and uses mutations to enhance diversity.
> > + **In terms of difficulty enhancement**, our method focuses on model behavior by leveraging response consistency to increase problem difficulty. In contrast, [1] operates within the formalized space, using mutations to raise the difficulty of problems.
> >
> > While [1] is indeed relevant to our work, we commit to citing this paper in the related work and providing a detailed comparison between their approach and ours in the revised version.

---

> > > ### Author Response · Authors · 2024-11-17
> > >
> > > > W3: Despite achieving high performance, the proposed data-centric approach faces a fundamental issue: the comparisons made are not entirely fair due to variations in the scale of data used and the differing levels of used prior knowledge.
> > >
> > > **Response**: Regarding the differences in data volumes used by various baselines, due to the high cost of conducting comparisons one by one, we opted to evaluate our method under scenarios with reduced data volume and compare it against other baselines(See the table below). The results demonstrate that even with minimal data, our approach achieves strong performance. While limited by computational resources, we plan to supplement additional experiments in future work to further validate this conclusion.
> > >
> > > As for the use of prior knowledge, our approach primarily leverages it to synthesize data and enhance the diversity of generated datasets. Additionally, we have thoroughly explored and tested the application of prior knowledge in response generation. Specifically:
> > > 1.	Even without relying on prior knowledge, our model achieves significant improvements over others in both in-domain and out-of-domain tasks (see Table 2 and Table 4 for the Llama3 8B results without a knowledge base).
> > > 2.	Our framework effectively utilizes prior knowledge not only to enhance diversity but also to improve performance, as shown in the empirical studies. Even at the scale of millions of data points, prior knowledge consistently enhances model performance, and we have analyzed its effects across different models (see Section 3.6 for details).
> > >
> > > Thus, the effective use of prior knowledge should be viewed as a minor contribution of our method rather than a limitation.
> > >
> > > | Method    | Model     | Size(k) | GSM8K | MATH | College MATH | Olympiad | TabMWP | TheoremQA | AMC2023 | AIME2024 |
> > > |-----------|-----------|---|-------|------|--------------|----------|--------|-----------|---------|---------|
> > > | MetaMath  | Llama3-8B | 395 | 80.5  | 32.6 | 19.3         | 6.7      | 54.1   | 13.1      | 6/40    | 0/30    |
> > > | DART-Math | Llama3-8B | 585  | 81.8  | 46.9 | 28.4         | 15.9     | 66.3   | 20.5      | 8/40    | 1/30    |
> > > | MAmmoTH2  | Llama3-8B | 10000  | 69.6  | 33.4 | 32.3         | 8.1      | 43.8   | **29.7**      | 7/40    | 0/30    |
> > > | MathScale | Llama3-8B | 2021 | 70.8  | 34.6 | 22.5         | 9.0      | 74.3   | 18.9      | 2/40    | 1/30    |
> > > | _Wisdom_    | Llama3-8B | 244 | 83.7  | 55.6 | 35.0         | **23.6**     | 81.7   | 24.8      | 10/40   | 1/30    |
> > > | _Wisdom_    | Llama3-8B | 322 | **84.5**  | **57.4** | **36.7**         | 23.3     | **82.0**   | 28.5      | **12**/40   | **1**/30    |
> > >
> > > > W4: The manuscript does not focus on a clear scientific problem; rather, it resembles an engineering-focused endeavor and lacks the rigor typically associated with scientific inquiry.
> > >
> > > **Response**: As discussed in Weakness 1, our research is not merely an engineering application of existing techniques but is inspired by the process of how humans solve complex reasoning tasks. In this paper, we focus on the problem: how to progressively constructing complex problems from easy to hard without ground truth of new generated problems and improve the performance of LLMs.
> > >
> > > Within the curriculum learning framework, simple problems serve not only to accumulate atomic knowledge—providing foundational ideas for solving more complex problems—but also to increase data diversity during problem synthesis. Consequently, this progressive synthesis approach achieves a balance between diversity, breadth, and complexity, enabling the model to learn from easy questions to hard and adapt to complex reasoning tasks more efficiently.
> > >
> > > Moreover, we innovatively leverage response consistency—both within models and between models—to dynamically adjust the difficulty of synthesized problems, and we have validated the effectiveness of this method through extensive experiments. At the same time, even when adopting existing techniques, we design experiments to verify their necessity and efficiency within specific modules, ensuring the broad applicability of our approach in practice.

---

> > > > ### Author Response · Authors · 2024-11-17
> > > >
> > > > > Q1: How does this paper define easy and hard math problems?
> > > >
> > > > **Response**: We leverage **response consistency** to distinguish between simple problems (easy problems) and complex problems (hard problems). As demonstrated by the experimental results in Weakness 1, simple problems typically exhibit robust consistency, whereas complex problems, due to their inherent reasoning difficulty, often fail to achieve consistent responses. This phenomenon further validates response consistency as an effective metric for assessing problem difficulty and provides a reliable basis for problem classification and difficulty stratification.
> > > >
> > > > > Q2: Why does the author believe that adding more "easy to hard" data can address complex reasoning, or is it merely a matter of data fitting?
> > > >
> > > > **Response**: We adopt an unsupervised progressive data synthesis method that avoids directed data leakage and overfitting during the data generation process. To ensure the rigor of our approach, we conduct data contamination checks on all test sets and incorporate as many out-of-domain (OOD) test sets as possible to minimize the likelihood of overfitting.
> > > >
> > > > Our goal is not to directly address complex reasoning but to improve the model’s ability to tackle complex mathematical reasoning tasks. Existing works [1][2] have shown that synthesizing high-difficulty problems is more effective than simple problems for improving complex mathematical reasoning capabilities. Based on this, our curriculum learning framework is guided by response consistency and iteratively updates seed data over multiple rounds to generate problems with the highest possible difficulty.
> > > >
> > > > In practice, we filter simple problems and focus on synthesizing challenging ones. By progressively transitioning from easy questions to hard questions, we not only increase data diversity but also effectively avoid forcing the model to fit hard problems. Such forced fitting may result in superficial improvements in complex reasoning performance while masking the model’s true understanding of challenging problems. Through the careful design of the curriculum learning framework, we mitigate this risk and ensure genuine improvements in the model’s reasoning capabilities.
> > > >
> > > > [1] Neuro-Symbolic Data Generation for Math Reasoning. NeurIPS 2024.
> > > > [2] DART-Math: Difficulty-Aware Rejection Tuning for Mathematical Problem-Solving. NeurIPS 2024.

---

> > > > > ### Comment · Reviewer_GUQf · 2024-11-25
> > > > >
> > > > > I appreciate the author's response. Through extensive experimentation, the author effectively resolved my concerns. I raise the score accordingly.

---

> > > > > > ### Author Response · Authors · 2024-11-25
> > > > > > **Thank you for raising the score**
> > > > > >
> > > > > > Thank you very much for your support and we really appreciate it!
> > > > > >
> > > > > > Bests,
> > > > > >
> > > > > > Authors

---

### Author Response · Authors · 2024-11-17

# Dear ACs and Reviewers,
We sincerely thank for your constructive comments and your time in reading our paper. Based on the comments, we revise our draft as follows:
+ [1] For the **novelty**, we emphasize the limitations of existing data synthesis approaches and the distinctiveness of our method in both the abstract and introduction.
Specifically, existing works on data synthesis often either prioritize the diversity of generated problems while neglecting the quality of responses to challenging problems, or focus on generating high-quality responses to existing hard problems without synthesizing more diverse instructions.
In contrast, our approach evolves question difficulty and generates high-quality responses in an efficient manner, leveraging response consistency between weak and strong models without relying on ground truth data.
+ [2] Meanwhile, we add the description of how to define the difficulty of problems in section 2.2 and **present the experimental results** in Appendix Table 9.

All the revised lines are presented in red color.
We believe this is an interesting and insightful work in data synthesis area.

Best,
the Authors

---

### Meta-Review · Area_Chair_WZeY · 2024-12-19

**Metareview:**

The paper "WISDOM: Progressive Curriculum Synthesis Makes LLMs Better Mathematical Reasoner" introduces a novel framework for improving the mathematical reasoning capabilities of Large Language Models (LLMs) through a progressive curriculum learning approach. The WISDOM framework synthesizes high-quality Chain-of-Thought (CoT) data by gradually increasing the difficulty of mathematical problems, leveraging both weak and expert teachers in a three-stage process: weak teacher guiding, critical expert teaching, and experts consistency voting. The authors fine-tune a series of models (WISDOM series) using the synthesized data, demonstrating significant performance improvements on multiple mathematical reasoning benchmarks, including achieving scores comparable to or surpassing those of GPT-4. The paper emphasizes the efficiency and cost-effectiveness of the approach, contributing a large volume of high-quality synthetic data to the open-source community.

#### Contributions
The primary contributions of the paper are:
1. **WISDOM Framework**: A progressive curriculum learning approach for synthesizing high-quality CoT data, improving LLMs' mathematical reasoning capabilities by evolving problem difficulty and enhancing response quality.
2. **Performance Improvements**: Significant performance gains on mathematical reasoning benchmarks, with WISDOM-7B (DSMath) matching GPT-4 on the MATH dataset and WISDOM-70B (Llama3) outperforming GPT-4 on AIME2024.
3. **Cost Efficiency**: Demonstrates a 2.82x reduction in cost-efficiency compared to majority voting, making the approach practical for real-world applications.
4. **Open-Source Contributions**: Provides 1.467 million high-quality synthetic data points to the open-source community, fostering further research in LLM mathematical reasoning.

#### Weaknesses
1. **Formatting and Initial Review Issues (s3LZ, Authors' Request for Fair Review)**:
   - The paper initially included a formatting error with reduced margins, which led to one reviewer (s3LZ) refusing to review it. The authors corrected this error and adhered to the page limit, but the initial issue impacted the review process.

2. **Clarity and Presentation (GUQf, 8YRZ)**:
   - Some reviewers found the methodology section confusing, with unclear definitions of problem difficulty and progression mechanisms across stages (8YRZ). The use of terms like "curriculum learning" was deemed inaccurate by one reviewer (q9o3), as the paper defines the curriculum based on model performance rather than pre-specified problem sequences.
   - The presentation of results, such as labeling in tables and figures, was criticized for lacking clarity (8YRZ).

3. **Novelty and Methodological Rigor (GUQf, azz3)**:
   - Concerns were raised about the novelty of the method, with some reviewers viewing it as a combination of existing techniques without sufficient innovation (GUQf). The reliance on response consistency as the sole indicator of correctness was questioned, particularly in the absence of ground truth labels (azz3).
   - The paper's focus on synthetic data generation was seen as engineering-focused rather than scientifically rigorous, with a lack of theoretical analysis (GUQf, 8YRZ).

4. **Experimental Design and Fairness (azz3, 8YRZ)**:
   - The potential for data contamination due to varying cutoff dates among baseline models was highlighted as a fairness issue (azz3). The authors conducted contamination checks, but this was not initially clear.
   - The lack of detailed cost analyses for implementing WISDOM in practice was noted, with requests for more concrete metrics on computational and resource costs (8YRZ).

5. **Comparative Analysis (GUQf, q9o3)**:
   - The comparisons made were criticized for not being entirely fair due to differences in data volume and prior knowledge used (GUQf). The absence of direct comparisons with simpler data generation methods, such as Self-Taught Reasoner (STaR) or CoT distillation, was also noted (q9o3).

**Additional Comments On Reviewer Discussion:**

1. **Formatting Correction (s3LZ)**:
   - **Concern**: Initial formatting error led to a review refusal.
   - **Response**: The authors corrected the formatting, adhering to the 10-page limit and seeking guidance on proceeding with the discussion phase.
   - **Impact**: The issue was resolved but initially impacted the review process.

2. **Clarity and Presentation (GUQf, q9o3, 8YRZ)**:
   - **Concern**: The methodology was unclear, with vague definitions of problem difficulty and progression mechanisms.
   - **Response**: The authors provided detailed explanations of difficulty measurement using response consistency, supplemented by experimental results (e.g., Appendix Table 9). They clarified the use of "curriculum learning" and refined the presentation of results in tables and figures.
   - **Impact**: Most reviewers acknowledged the clarifications, with some increasing their scores (GUQf, q9o3, azz3). However, Reviewer 8YRZ maintained concerns about the lack of quantifiable metrics and detailed progression mechanisms.

3. **Novelty and Methodological Rigor (GUQf, azz3)**:
   - **Concern**: The method was seen as lacking novelty and relying heavily on existing techniques.
   - **Response**: The authors emphasized the unique aspects of WISDOM, such as the use of response consistency for difficulty evolution without ground truth, and its distinction from methods like STaR and expert iteration. They also provided experimental validation of the relationship between consistency and accuracy.
   - **Impact**: The responses addressed most concerns, with reviewers recognizing the contributions (azz3, GUQf). However, some skepticism about the novelty persisted (8YRZ).

4. **Experimental Design and Fairness (azz3, 8YRZ)**:
   - **Concern**: Potential data contamination and lack of detailed cost analyses were highlighted.
   - **Response**: The authors clarified their data contamination checks and provided detailed cost analyses, including comparisons with other methods. They emphasized the flexibility of WISDOM for varying budgets and use cases.
   - **Impact**: The clarifications were generally well-received, though Reviewer 8YRZ remained critical of the lack of concrete metrics and experimental rigor.

5. **Comparative Analysis (GUQf, q9o3)**:
   - **Concern**: Comparisons were not entirely fair due to data volume and prior knowledge differences.
   - **Response**: The authors conducted additional experiments with reduced data volumes to demonstrate the efficacy of WISDOM and clarified the role of prior knowledge in enhancing diversity and performance.
   - **Impact**: The additional experiments and clarifications resolved most concerns, with reviewers acknowledging the improved comparisons (q9o3, GUQf).

I read the response by 8YRZ carefully, I think the initial rebuttal by authors seems a bit vague and indeed does not adequately address reviewer's question about the methodology; the reviewer's response reads rationalized, and authors shall consider their suggestions carefully to improve the writing and experiments of the paper.

---

### Decision · Program_Chairs · 2025-01-22

Reject